# BIOMEDSQL: TEXT-TO-SQL FOR SCIENTIFIC REASONING ON BIOMEDICAL KNOWLEDGE BASES

## ABSTRACT

Biomedical researchers increasingly rely on large-scale structured databases for complex analytical tasks. However, current text-to-SQL systems often struggle to map qualitative scientific questions into executable SQL, particularly when implicit domain reasoning is required. We introduce *BiomedSQL*, the first benchmark explicitly designed to evaluate scientific reasoning in text-to-SQL generation over a real-world biomedical knowledge base. BiomedSQL comprises 68,000 question/SQL query/answer triples generated from templates and grounded in a harmonized BigQuery knowledge base that integrates gene–disease associations, causal inference from omics data, and drug approval records. Each question requires models to infer domain-specific criteria, such as genome-wide significance thresholds, effect directionality, or trial phase filtering, rather than rely on syntactic translation alone. We evaluate a range of open- and closed-source LLMs across prompting strategies and interaction paradigms. Our results reveal a substantial performance gap: GPT-o3-mini achieves 59.0% execution accuracy, while our custom multi-step agent, BMSQL, reaches 62.6%, both well below the expert baseline of 90.0%. BiomedSQL provides a new foundation for advancing text-to-SQL systems capable of supporting scientific discovery through robust reasoning over structured biomedical knowledge bases.

## 1 INTRODUCTION

Modern biomedical research is increasingly data-centric. Electronic health records (EHRs), high-throughput assays, and population-scale studies populate large databases that researchers query daily. Natural-language interfaces, particularly text-to-SQL systems, offer the promise of democratizing access to these resources. However, most current systems treat query generation as a syntactic translation task, mapping question structure to SQL templates without deeper domain understanding.

This abstraction breaks down in biomedical contexts. Domain experts routinely ask questions such as *"What SNPs are most significantly associated with Alzheimer's disease?"* or *"What drugs target genes up-regulated in Parkinson's disease?"*, questions grounded in implicit scientific conventions, such as statistical thresholds, drug approval pathways, and causal inference across multiple modalities.

These domain-specific conventions, e.g., genome-wide significance cutoffs, trial phase filtering, or effect-size interpretation, are invisible from the schema alone. While general-purpose text-to-SQL benchmarks (e.g., SPIDER (Yu et al., 2018), BIRD (Li et al., 2023)) have advanced the field, they do not evaluate the scientific reasoning required in complex domains. Similarly, EHR-focused benchmarks (Wang et al., 2020a; Lee et al., 2022) emphasize temporal logic or patient retrieval, but do not isolate or rigorously test **scientific reasoning on large-scale databases** that is required for interpreting biomedical data. This includes inferring statistical significance thresholds or chaining multi-step filtering logic across ontologies the way a skilled biomedical analyst would.

To address this gap, we introduce **BiomedSQL**, the first benchmark specifically designed to evaluate **scientific reasoning in SQL generation** within the biomedical domain. BiomedSQL contains 68,000 biomedical question/SQL query/answer triples that reflect realistic, complex scientific queries. These queries were templated from a set of 40 unique questions that were generated and verified by domain experts for their real-world biomedical research impact and ability to be answered through the use of structured data. These are executable against a harmonized, publicly available BigQuery database integrating gene–disease associations, multi-omic causal inferences, and drug approval records. These

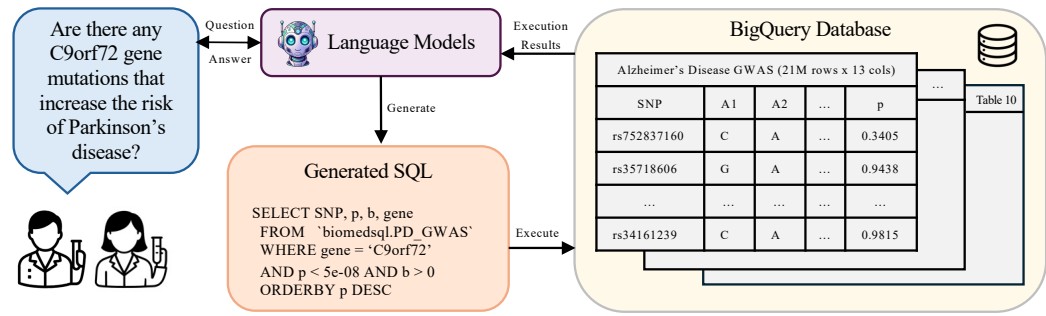

Figure 1: Example text-to-SQL workflow used to evaluate LLM performance on BiomedSQL. Given a question and the database schema information, an LLM must generate a SQL query and use its execution results to return a natural language response.

data sources were also verified as being the most up-to-date by domain experts that use them in their day-to-day computational research activities. Each question in BiomedSQL requires models to translate nuanced, qualitative scientific language into precise, executable SQL logic.

Figure 1 provides an overview of our evaluation pipeline, where we task various LLMs in a variety of prompting and interaction scenarios on their ability to generate accurate SQL queries and summarize the execution results into a concise, natural language response based on the provided question. Our evaluation on BiomedSQL indicates significant room for improvement in deploying LLMs on text-to-SQL workflows for biomedical knowledge bases. The top-performing model, GPT-o3-mini, only achieves an execution accuracy of 59.0%, even when provided with sample table rows, example queries, and explicit domain-specific instructions (e.g., significance thresholds). Our custom multi-step agent, BMSQL, improves this to 62.6%, but still trails expert performance (90%).

**Our contributions are as follows:**

- **Benchmark:** We introduce BiomedSQL, a benchmark of 68,000 augmented question/SQL query/answer triples designed to evaluate scientific reasoning capabilities in text-to-SQL systems on a realistic, multi-table biomedical knowledge base.
- **Infrastructure:** We release a harmonized BigQuery database, expert-authored gold queries, execution results, and a toolkit for performance evaluation and agent orchestration.
- **Evaluation:** We assess a range of models, prompting styles, and multi-step interaction paradigms, including a custom-built text-to-SQL system (BMSQL), revealing a 30-40% gap compared to domain expert-level performance.

## 2 RELATED WORK

We organize related work into four strands: (1) general-purpose text-to-SQL benchmarks, (2) text-to-SQL benchmarks for clinical databases, (3) evaluations of scientific reasoning in NLP, and (4) a direct comparison between BiomedSQL and previous benchmarks.

**General-purpose text-to-SQL benchmarks:** Text-to-SQL research has been largely driven by cross-domain benchmarks. Early work such as Seq2SQL (Zhong et al., 2017) introduced SQL generation for simple, single-table queries. SPIDER (Yu et al., 2018) expanded generalization challenges by spanning 200 multi-table databases, catalyzing the development of schema linking techniques. More recently, KaggleDBQA (Lee et al., 2021) and BIRD (Li et al., 2023) added realism by incorporating enterprise-scale data and requiring attention to data quality, joins, and latency. Despite progress, top models still underperform humans by 15-20% on execution accuracy for general-purpose benchmarks like BIRD. These benchmarks emphasize syntactic translation and schema generalization, not the domain-specific reasoning that underlies many queries in scientific disciplines. In contrast, BiomedSQL uses a multi-table biomedical schema enriched with domain-specific semantics, requiring models to interpret statistical conventions and biomedical ontologies—capabilities not previously assessed.

**Text-to-SQL benchmarks for clinical databases:** Several efforts have targeted clinical database querying for EHRs. MIMICSQL (Wang et al., 2020b) presented a synthetic SQL benchmark over MIMIC-III but was limited by its narrow schema. EHRSQL (Lee et al., 2022) advanced realism by crowdsourcing 1,000+ natural language queries from clinicians across MIMIC-III and eICU, highlighting challenges in temporal logic, answerability, and data sparsity. Recent datasets have diversified query paradigms across relational, document, and graph data models (Sivasubramaniam et al., 2024), reflecting the increasing heterogeneity of EHR data. These benchmarks focus on patient-centric information retrieval and assess the ability to map schema-dependent queries. Conversely, BiomedSQL targets scientific discovery in biomedical research, where queries require implicit scientific reasoning such as applying significance thresholds, combining multi-omic associations, or resolving drug–gene–disease relationships. This emphasis offers a complementary evaluation to clinical datasets by focusing on reasoning-heavy data exploration.

**Evaluating scientific reasoning in NLP:** Scientific reasoning has emerged as a critical frontier in NLP for tasks requiring multi-hop inference, evidence synthesis, and structured decision-making. Benchmarks such as SciFact (Wadden et al., 2020) and EntailmentBank (Dalvi et al., 2021) evaluate scientific claim verification and multi-step reasoning over textual evidence. Prompting techniques like Chain-of-Thought (Wei et al., 2022) and ReAct (Yao et al., 2023a) have demonstrated improved performance on multi-step reasoning tasks in both general and biomedical settings. More recent efforts have extended evaluation to structured data, such as TabularBench (Wang et al., 2025), Hierarchography (Gao et al., 2025), and SQL-RL (Ma et al., 2025), which assess LLM reasoning over tables, ontologies, and relational programs. Despite these advances, critical challenges remain in aligning model reasoning with biomedical standards of rigor, safety, and explainability. BiomedSQL addresses this gap by evaluating models on their ability to infer and operationalize scientific reasoning, including statistical thresholds, ontology resolution, and complex filtering, in text-to-SQL generation over large-scale biomedical knowledge bases.

**Benchmark comparison.** We compare BiomedSQL to several recent text-to-SQL benchmarks in Table 1. BiomedSQL is unique in four ways: (1) it contains the largest number of question/SQL/answer triples, (2) it features some of the longest average SQL queries (second only to EHRSQL), (3) it explicitly targets scientific reasoning rather than schema translation, and (4) it evaluates both the generated SQL and the model's natural language response. BiomedSQL is also the only biomedical domain-specific benchmark that tests the model's ability to utilize knowledge outside of the scope of the schema itself. Additionally, it is the only benchmark using BigQuery, a cloud-native SQL dialect, thus further simulating deployment-relevant environments.

Table 1: Comparison of BiomedSQL to other text-to-SQL benchmarks. BiomedSQL uniquely evaluates scientific reasoning and natural language responses while supporting BigQuery execution.

| Dataset | Number of Questions | Number of Queries | Avg. Tokens | Knowledge | Template | Scientific Reasoning | NL Response | BigQuery |
|---|---|---|---|---|---|---|---|---|
| MIMICSQL (Wang et al., 2020b) | 10,000 | 10,000 | 57.4 | ✗ | ✓ | ✗ | ✗ | ✗ |
| EHRSQL (Lee et al., 2022) | 24,000 | 24,000 | **109.9** | ✗ | ✓ | ✗ | ✗ | ✗ |
| SM3 (Sivasubramaniam et al., 2024) | 10,000 | 40,000 | 26.1 | ✗ | ✓ | ✗ | ✗ | ✗ |
| BIRD (Zhang et al., 2023) | 12,751 | 12,751 | 50.6 | ✓ | ✗ | ✗ | ✗ | ✗ |
| **BiomedSQL** | **68,227** | **68,227** | 96.4 | ✓ | ✓ | ✓ | ✓ | ✓ |

## 3 BIOMEDSQL CONSTRUCTION

BiomedSQL is designed to evaluate scientific reasoning in text-to-SQL generation over structured biomedical knowledge. We construct it by: (1) harmonizing a multi-source relational database to support biomedical queries, (2) authoring gold-standard SQL annotations from domain experts, and (3) augmenting the dataset from templates to produce 68,000 question/SQL query/answer triples.

### 3.1 RELATIONAL DATABASE CONSTRUCTION

We first constructed a relational database that spans ten core tables drawn from trusted biomedical resources, ensuring sufficient coverage for answering the full set of 68,000 questions. Data was pre-processed for consistency and deployed to BigQuery for efficient querying and public reproducibility.

Our primary data sources include the **OpenTargets Platform** (Targets, 2024), which aggregates gene–disease–drug associations, and **ChEMBL** (Méndez et al., 2024), a manually curated database of bioactive molecules and pharmacological data. OpenTargets data was retrieved via FTP and cleaned manually, while ChEMBL data was accessed through Google BigQuery and normalized by flattening nested fields. Together, these sources provide a unified schema of gene–disease links, drug–target pairs, trial status, and pharmacodynamics.

To support questions involving statistical genetics, we included summary statistics from large-scale GWAS studies of Alzheimer's disease (Bellenguez et al., 2022) and Parkinson's disease (Nalls et al., 2019), obtained from the GWAS Catalog. We retained SNP-level data including p-values, rsIDs, allele frequencies, and nearest-gene mappings after quality control filtering.

We integrated causal inference data from **omicSynth** (Alvarado et al., 2024), which applies summary-data-based Mendelian randomization (SMR) to identify multiomic biomarkers with putative causal links to neurodegenerative diseases. These datasets enable reasoning over associations not directly stated but statistically inferred, e.g. *"What metabolites causally influence Parkinson's progression?"*

All tables were normalized and uploaded in Parquet format to Google Cloud BigQuery. A full schema and column listing are provided in Appendix A.1. To support future expansions, we have curated additional tables that extend BiomedSQL's coverage to broader omics and clinical-trial data.

## 3.2 SQL Annotation and Augmentation

**Gold SQL authoring.** To ensure executable grounding, a domain expert manually wrote gold-standard SQL queries for each of the 40 seed questions drawn from CARDBiomedBench (Bianchi et al., 2025). Each query was crafted to retrieve the minimum evidence necessary to answer the question, avoiding SELECT * patterns and capping results at 100 rows. Two additional analysts reviewed all queries for syntactic correctness and semantic fidelity.

**Programmatic scaling.** Each of the 40 queries was then templated and automatically expanded using entity substitution. We aligned these templates to the full set of 68,000 QA pairs in CARDBiomed-Bench by programmatically inserting disease, gene, SNP, and compound mentions into the query templates. All generated SQL queries were executed on the BigQuery database to obtain execution results, which serve as ground-truth evidence for evaluating LLM-generated SQL queries.

This pipeline produced a benchmark where each QA pair is linked to an executable SQL query and its result. This enables precise evaluation of models' ability to translate scientific questions into domain-grounded, semantically valid, and executable SQL logic.

## 4 Dataset Analysis

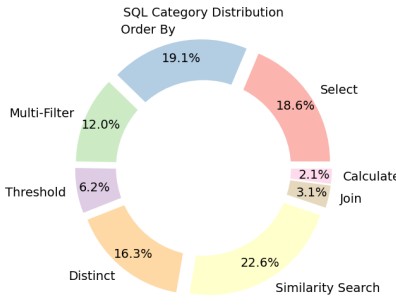

Figure 2: SQL category distribution.

**Task distribution and SQL complexity.** To characterize the scientific and computational complexity of BiomedSQL, we annotated all 68,000 question–query pairs with SQL operation types and biomedical reasoning categories. Table 2 defines the SQL operation categories, and Figure 2 shows their empirical distribution. Simpler operations—such as *Select*, *Order-By*, and *Calculate*—require relatively shallow syntactic parsing, which LLMs tend to perform well on. In contrast, operations such as *Multi-Filter*, *Threshold*, *Join*, and *Similarity Search* present greater difficulty, as they demand multi-step logical composition, implicit schema linking, or pattern-based retrieval.

**Scientific reasoning categories.** To probe scientific reasoning, we classified BiomedSQL queries into three reasoning categories reflecting cognitive processes typical of biomedical experts:

1. **Operationalizing implicit scientific conventions:** Queries often invoke domain-specific concepts (e.g., "significantly associated SNPs") that imply non-obvious statistical thresholds, such as $p < 5 \times 10^{-8}$ for GWAS hits or directionality based on beta coefficients. These conventions are rarely explicit in schemas and must be inferred by models.

Table 2: Description of SQL query categories in BiomedSQL.

| SQL Category | Description |
|---|---|
| Select | Retrieves specific columns from one or more tables. |
| Distinct | Retrieves unique values from specified columns. |
| Join | Combines rows across multiple tables via relational keys. |
| Multi-Filter | Applies compound filtering logic (AND, OR, NOT). |
| Threshold | Filters based on logical or statistical thresholds (e.g., p-values). |
| Calculate | Performs arithmetic operations (e.g., counts, averages). |
| Order-By | Sorts the result set by specified columns. |
| Similarity Search | Performs pattern-based retrieval using LIKE, regex, or full-text search. |

2. **Incorporating missing contextual knowledge:** Experts often incorporate auxiliary data (e.g., drug approval status or clinical trial phase) even when not directly mentioned. For instance, determining if a drug is "approved" for a condition requires disambiguating indication-specific trial phase information beyond any binary "approved" flag.

3. **Executing complex multi-hop reasoning workflows:** Many questions in BiomedSQL require chaining relational operations across multiple tables. For example, "Which tissues are genes associated with Parkinson's disease most significantly expressed in?" requires a four-step inference over gene–disease, gene–expression, tissue annotations, and statistical ranking. LLMs often struggle to translate such multi-hop logic into valid, executable SQL.

Additional biological reasoning categories and visualization of their distribution are provided in Appendix A.2 (Table 6, Figure 4).

## 5 Experiments

The goal of our experiments is to assess how well LLMs can translate biomedical natural language questions from BiomedSQL into accurate and executable BigQuery SQL queries. We evaluate models across different interaction paradigms and prompting configurations on a test set of 546 questions, comparing SQL execution accuracy and the quality of the natural language answers they generate.

### 5.1 Experimental Setup

**Models.** We evaluate a range of state-of-the-art open-source and proprietary LLMs. Open-source models include **LLaMA-3.1** (70B, 405B) and **Qwen-2.5-Coder** (14B, 32B). Closed-source models include the **GPT** family (GPT-4o, GPT-4o-mini, GPT-o3-mini), the **Gemini-2.0-flash** family (flash and flash-lite), and **Claude-3.7-sonnet**. This selection spans a diverse range of parameter scales, computational cost profiles, and architectural design philosophies.

**Isolated SQL generation.** We first assess models in a single-turn text-to-SQL setting. Each model receives a baseline prompt containing: (1) the natural language biomedical question from the benchmark, (2) the database schema describing tables, columns, and relationships, and (3) simple instructions on how to structure the SQL query and remain consistent with the database schema.

To study prompt sensitivity, we vary prompt structure along several axes: (1) adding sample table rows (*3-rows*, *5-rows*), (2) adding few-shot examples (*1-shot*, *3-shot*, *5-shot*, *10-shot, 20-shot, 40-shot*), (3) adding explicit domain-specific instructions (e.g., statistical thresholds via *stat-instruct*), and (4) a combined variant that includes *3-rows*, *3-shot*, and *stat-instruct* (*combo*). Prompt templates are provided in Appendix A.3.

**Interaction paradigms.** Beyond single-turn prompts, we investigate multi-step paradigms that allow iterative reasoning and query refinement. These systems can request schema details, propose intermediate queries, and update their approach before presenting a final SQL query. These paradigms were evaluated using GPT and Gemini models. We experiment with four architectural variants:

1. **ReAct**: A prompt-orchestrated approach where schema validation, syntax checking, and other external tools are invoked within multi-step SQL generation steps (Yao et al., 2023b). The react prompts used are detailed in Appendix A.4.

2. **Schema Indexing**: Schema descriptions are dynamically retrieved using LlamaIndex to support contextual grounding and table selection.
3. **DAIL-SQL**: We adapt DAIL-SQL (Gao et al., 2023) for use on BiomedSQL. DAIL-SQL is a state-of-the-art text-to-SQL solution that retireves relevant example SQL queries based on the question and injects them into the prompt for more accurate query generation. It is consistently near the top of leaderboards for popular benchmarks like SPIDER and BIRD.
4. **Multi-step query refinement**: We implement an iterative text-to-SQL architecture, called BMSQL, where an initial query is refined through feedback loops based on intermediate results or execution errors, emulating expert-level refinement processes. The implementation details of BMSQL are shown in Appendix A.5.

**SQL execution metrics.** We report three SQL performance metrics: **Execution Accuracy (EX)**, **Jaccard Index (JAC)**, and **Syntax Error Rate (SER)**. Execution Accuracy is a widely used text-to-SQL metric (Yu et al., 2018; Li et al., 2023) which represents the proportion of questions in the evaluation set for which the LLM-generated query and the ground-truth query return identical results. We adapt EX for our use case to measure row-wise set equality, comparing the set of UUIDs returned in the case of `SELECT *` queries or the set of numeric values returned in the case of `COUNT` and other calculation queries. Jaccard Index (Costa, 2021), or intersection over union, is a metric for gauging the similarity of two sets. It tells us how close the LLM-generated SQL query results are to the ground-truth. Unlike EX, JAC will still credit a query that returns slightly more or less rows than the ground-truth, making it a more lenient metric. Finally, Syntax Error Rate is simply the proportion of questions in the evaluation set for which the LLM-generated SQL query was not executable.

**BioScore.** To evaluate natural language responses, we adopt BioScore (Bianchi et al., 2025), an LLM-as-a-judge metric computed using GPT-4o. BioScore includes:

– **Response Quality Rate (RQR):** Proportion of factually correct answers. Measures how often a model provides correct answers.
– **Safety Rate (SR):** Proportion of abstentions among all incorrect or abstained answers. Assesses a model's ability to abstain from answering when uncertain.

All metric definitions and prompts are provided in Appendix A.6. A correlation analysis between the SQL execution and BioScore metrics is presented in Appendix A.7. To mitigate concern over the use of LLM-as-a-judge metrics, a correlation analysis between LLM-generated and domain expert-generated BioScores is presented in Appendix A.8. 100 LLM-generated responses were sampled from a variety of different tested models and interaction paradigms. Domain experts were then asked to grade these responses, and a comparison between the counts of the LLM-generated and domain expert-generated BioScores is presented in Table 7. The resulting Spearman correlation coefficient from these counts is 0.89 (p < 1e-5). This high correlation gives us confidence that the LLM-as-a-judge metrics used in our evaluations are stable and accurate.

**Domain Expert Baseline.** Two expert biomedical analysts answered a quiz over a representative sample of questions. For each, they generated SQL queries, execution results, and natural language answers. We report mean EX, JAC, and RQR. SR and SER are not available for this format, as experts could not abstain and produced valid SQL in all cases.

## 5.2 EVALUATION RESULTS

**LLMs struggle with scientific reasoning in SQL generation.** Table 3 shows that even top-performing models such as GPT-o3-mini (EX = 53.5%, JAC = 60.4%, RQR = 73.3%) fall short of domain expert performance (90–95%). GPT-4o performs slightly worse (EX = 46.9%, JAC = 54.7%, RQR = 71.2%). Among open-weight models, despite their small size, Qwen-2.5-Coder-32B achieves competitive EX (40.8%) and Qwen-14B attains strong RQR (62.1%), outperforming Llama models that dwarf them in terms of parameters. Claude-3.7-sonnet exhibits the best SR (43.0%), indicating better abstention behavior.

**Few-shot examples provide performance gains.** Table 8 in Appendix A.9 shows that the *10-shot* prompt yields the best improvement for GPT-o3-mini ($\Delta$EX=+7.8%, $\Delta$JAC=+7.3%, $\Delta$RQR=+7.0%). Interestingly, passing the model more example queries beyond 10, including the full set of 40 template queries, had little effect on performance. Passing raw table rows alone showed negligible benefit, underscoring that schema-level understanding matters more than content memorization.

Table 3: State-of-the-art LLMs struggle with scientific reasoning-based text-to-SQL tasks (*Domain expert baselines not available for SR, SER, and token count as described in §5.1).

| Model | EX (%) ↑ | JAC (%) ↑ | RQR (%) ↑ | SR (%) ↑ | SER (%)↓ | # Tokens |
|---|---|---|---|---|---|---|
| Domain Expert | 90.0 | 90.0 | 95.0 | NA* | NA* | NA* |
| GPT-4o | 46.9 (±4.2) | 54.7 (±3.8) | 71.2 (±3.8) | 26.1 (±3.7) | 1.3 (±0.9) | 3,689 |
| GPT-4o-mini | 35.9 (±4.0) | 41.4 (±3.9) | 60.6 (±4.1) | 23.3 (±3.5) | 11.0 (±2.6) | 89,612 |
| GPT-o3-mini | **53.5 (±4.2)** | **60.4 (±3.8)** | **73.3 (±3.7)** | 29.4 (±3.8) | **0.2 (±0.4)** | 3,942 |
| Gemini-2.0-flash | 33.7 (±4.0) | 37.0 (±3.9) | 71.1 (±3.8) | 27.2 (±3.7) | 4.2 (±1.7) | 3,692 |
| Gemini-2.0-flash-lite | 17.9 (±3.2) | 18.1 (±3.2) | 41.0 (±4.1) | 26.4 (±3.7) | 8.4 (±2.3) | 3,280 |
| Claude-3.7-sonnet | 45.4 (±4.2) | 49.8 (±4.0) | 69.8 (±3.8) | **43.0 (±4.1)** | 1.6 (±1.1) | 3,805 |
| Qwen-2.5-Coder-14B | 37.0 (±4.0) | 32.4 (±3.9) | 62.1 (±4.1) | 42.5 (±4.1) | 11.0 (±2.6) | 3,453 |
| Qwen-2.5-Coder-32B | 40.8 (±4.1) | 44.4 (±4.0) | 58.2 (±4.1) | 61.0 (±4.1) | 15.7 (±3.1) | 3,612 |
| Llama-3.1-70B | 34.4 (±4.0) | 39.8 (±3.9) | 57.0 (±4.1) | 37.0 (±4.0) | 6.0 (±2.0) | 3,547 |
| Llama-3.1-405B | 38.1 (±4.1) | 42.5 (±4.0) | 57.9 (±4.1) | 41.7 (±4.1) | 4.6 (±1.7) | 3,456 |

**Interaction paradigms yield mixed results.** Table 4 shows that schema indexing underperforms in both EX and RQR, likely due to its use of simple table descriptions and lightweight grounding. However, it exhibits the best SR (e.g., Index-GPT-4o = 66.9%), indicating it effectively abstains when uncertain. ReAct marginally improves EX for GPT variants but does not perform consistently across models. This suggests that ReAct-style prompts may need to be tuned to optimize performance on different models. Also, its high token usage makes it less practical for deployment. DAIL-SQL shows strong performance, rivaling that of BMSQL on all three models tested. However, it is important to note that even this state-of-the-art text-to-SQL approach still trails expert-level performance by 30%.

**BMSQL is the strongest performer overall.** Our custom system, BMSQL, outperforms all baselines. GPT-o3-mini with BMSQL achieves 62.6% EX and 69.2% JAC—both best in class. Paired with Gemini, BMSQL reaches 84.6% RQR, rivaling even domain experts on answer quality. However, execution accuracy remains significantly lower than expert benchmarks. These results highlight the value of domain-specific multi-step agents in structured biomedical tasks.

Table 4: Complex interaction paradigms provide mixed performance (*Gemini-2.0-flash is the Gemini model used for these experiments).

| Model | EX (%) ↑ | JAC (%) ↑ | RQR (%)↑ | SR (%) ↑ | SER (%)↓ | # Tokens |
|---|---|---|---|---|---|---|
| ReAct-GPT-4o | 49.6 (±4.2) | 57.9 (±3.8) | 67.2 (±3.9) | 8.9 (±2.4) | 0.0 (±0.0) | 14,286 |
| ReAct-GPT-o3-mini | 56.2 (±4.2) | 64.8 (±3.6) | 73.6 (±3.7) | 13.2 (±2.8) | 0.0 (±0.0) | 13,317 |
| ReAct-Gemini* | 48.9 (±4.2) | 56.6 (±3.8) | 60.4 (±4.1) | 10.2 (±2.5) | **0.0 (±0.0)** | 13,205 |
| Index-GPT-4o | 25.5 (±3.6) | 28.3 (±3.6) | 44.1 (±4.2) | **66.9 (±3.9)** | 27.5 (±3.7) | 1,110 |
| Index-GPT-o3-mini | 27.1 (±3.7) | 30.6 (±3.7) | 44.1 (±4.1) | 47.5 (±4.2) | 2.0 (±0.1) | 1,899 |
| Index-Gemini* | 46.1 (±4.2) | 48.5 (±4.1) | 54.2 (±4.2) | 59.6 (±4.1) | 8.1 (±2.3) | 787 |
| DAIL-SQL-GPT-4o | 54.8 (±4.2) | 58.1 (±3.4) | 75.5 (±3.4) | 63.4 (±4.0) | 6.6 (±2.1) | 3,624 |
| DAIL-SQL-GPT-o3-mini | 61.2 (±4.1) | 63.6 (±4.0) | 81.4 (±3.3) | 42.1 (±4.1) | 0.0 (±0.0) | 3,318 |
| DAIL-SQL-Gemini* | 53.1 (±4.2) | 58.8 (±3.4) | 82.8 (±3.1) | 30.6 (±3.7) | 0.0 (±0.0) | 3,185 |
| BMSQL-GPT-4o | 60.4 (±4.1) | 67.2 (±3.6) | 79.8 (±3.4) | 64.5 (±4.0) | 4.9 (±1.8) | 32,819 |
| BMSQL-GPT-o3-mini | **62.6 (±4.1)** | **69.2 (±3.6)** | 83.1 (±3.1) | 38.0 (±4.1) | 2.6 (±1.2) | 39,470 |
| BMSQL-Gemini* | 55.9 (±4.2) | 61.3 (±3.9) | **84.6 (±3.0)** | 32.1 (±3.9) | 0.2 (±0.4) | 22,045 |

# 6 ANALYSIS

To better understand model behavior beyond aggregate metrics, we analyze performance across SQL task types and evaluate the effects of increased inference-time compute.

**Performance across SQL categories.** Figure 3 presents radar plots showing the distribution of Execution Accuracy (EX) and Response Quality Rate (RQR) across SQL categories, as defined in §4. We evaluate GPT-o3-mini across four settings: (1) baseline prompt, (2) *combo* prompt, (3) ReAct prompting, and (4) BMSQL.

For EX, performance across SQL categories remains relatively stable across prompting strategies. As anticipated, models struggle most with *Join*, *Similarity Search*, and *Multi-Filter* queries, which require multi-table reasoning, implicit filtering, or fuzzy matching. Surprisingly, *Select* queries also show mid-range performance; however, this category includes a large portion of questions in BiomedSQL, so mean-level performance is expected. For RQR, BMSQL exhibits the most balanced performance across categories, likely due to its ability to: (1) apply domain-specific instructions (e.g., p-value thresholds, trial status), (2) compare thresholded vs. unthresholded results, (3) refine queries via execution feedback. This reinforces the benefits of multi-step pipelines in biomedical reasoning.

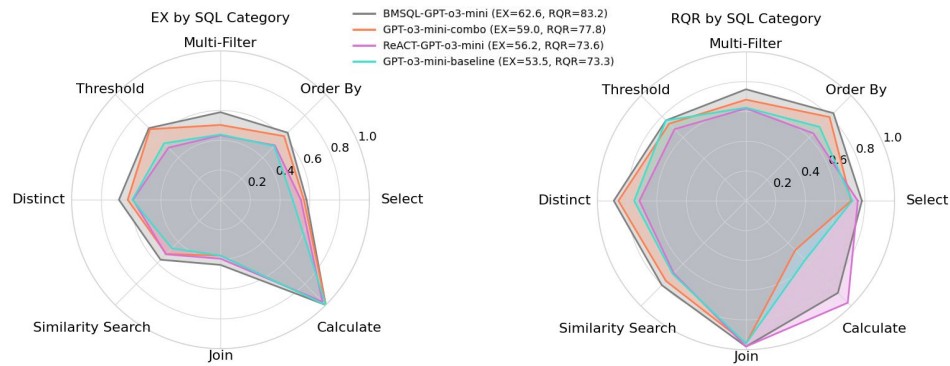

Figure 3: Distribution of performance across SQL categories for GPT-o3-mini in terms of EX (left) and RQR (right), across four prompting and interaction paradigms.

**Common SQL errors.** In order to determine the kinds of mistakes the LLMs are making in generating the SQL queries, we defined the following five error categories: (1) incorrect tables, (2) missing threshold, (3) incorrect threshold, (4) incorrect aggregations, and (5) syntax error. More precise definitions of each error category are presented in Appendix A.10, along with the counts of each error type for six of the top-performing models in Table 9.

Incorrect table selection is the most common error in the LLM-generated SQL queries, followed by missing or incorrect application of statistical thresholds. These results speak to the high correlation between the tables of the benchmark database and the lack of biomedical domain-specific reasoning ability by frontier LLMs, even when correct statistical thresholds are provided directly to the model.

**Effect of inference-time compute.** We next evaluate how model performance changes when given the opportunity to reflect and revise outputs over multiple reasoning steps. Specifically, we allow BMSQL-GPT-o3-mini to examine its initial SQL query, execution result, and answer, then choose to perform up to two additional passes if the outputs appear insufficient.

Results in Table 5 show that increasing inference steps yields marginal gains. From 1-pass to 3-pass, EX remains flat ($62.6\% \rightarrow 61.7\%$), while RQR increases slightly ($83.1\% \rightarrow 85.5\%$). Improvements in SR and elimination of syntax errors (SER = 0.0%) suggest that most corrections involve fixing syntax or abstaining rather than reformulating the query logic. Notably, BMSQL rarely chooses to invoke a third pass, as reflected in the minor token increase between the 2-pass and 3-pass settings.

Table 5: Increased inference time compute has little effect on performance of BMSQL-GPT-o3-mini.

| Model | EX (%) ↑ | JAC (%) ↑ | RQR (%) ↑ | SR (%) ↑ | SER (%) ↓ | # Tokens |
|---|---|---|---|---|---|---|
| 1-pass | **62.6** (±4.1) | 69.2 (±3.6) | 83.1 (±3.1) | **38.0** (±4.1) | 2.6 (±1.2) | 39,470 |
| 2-pass | 62.1 (±4.1) | **69.4** (±3.5) | 84.2 (±3.1) | 29.1 (±3.8) | **0.0** (±0.0) | 53,773 |
| 3-pass | 61.7 (±4.1) | 69.2 (±3.5) | **85.5** (±2.9) | 36.7 (±4.0) | 0.0 (±0.0) | 55,948 |

**Effect of database size.** BiomedSQL was created from a database of ten tables. While the count is small compared to other benchmarks, the size of the tables is much larger, spanning up to 72 million rows and 30 columns. This presents the LLMs with the difficult task of finding the correct columns to use and applying the proper thresholds based on the semantics of the presented question. The smaller table count is also justified by model performance on the benchmark: even the top performers trail expert-level execution accuracy by nearly 30%.

To thoroughly elucidate model performance on BiomedSQL, we added ten new tables to our database, including additional GWAS results from different diseases, drug mechanism of action data, and population-level allele frequencies. We evaluate GPT-o3-mini across three settings on this larger schema: (1) baseline prompt, (2) *combo* prompt, and (3) BMSQL. Table 10 in Appendix A.11 shows the results from this analysis. For the baseline and combo experiments, the drop in performance for EX, JAC, and RQR when moving to the larger schema ranges from 7-10%. The performance drop seen by BMSQL is less dramatic, ranging from 2-4% depending on the metric. These experiments show that, as expected, adding more tables to the database increases the difficulty of the text-to-SQL generation task for state-of-the-art models. We also discuss future benchmark expansion plans in §7.

## 7 DISCUSSION AND LIMITATIONS

**Use of template questions.** BiomedSQL was constructed from a set of 40 template questions. While templating can cause homogeneous samples compared to a real-world setting, it is common in the text-to-SQL community (Gao et al., 2023; Gan et al., 2021; Saparina and Lapata, 2024). Popular text-to-SQL benchmarks with large or no template sets often crowdsource their tasks (e.g., BIRD (Li et al., 2023)), which **only works when questions come from general domains that crowdworkers can reliably annotate** (i.e. sports, movies, and sales). Conversely, **BiomedSQL covers a highly technical domain that cannot be easily crowdsourced**. Instead, all questions and SQL queries were authored by domain experts (co-authors of this work) who are advanced biomedical data scientists and regularly implement complex bioinformatics workflows. Their expertise is essential for generating valid, technically meaningful text-to-SQL annotations, and the resulting dataset required a careful, detail-oriented construction process.

Recent work further demonstrates that templating does not trivialize evaluation. In GSM-Symbolic (Mirzadeh et al., 2025), regenerating questions through simple templates (e.g., replacing only surface-level entities such as names or numeric values) still causes LLMs to exhibit significant accuracy drops and increased performance variance relative to the original benchmark, even though the underlying question structure and required reasoning remain unchanged. These results indicate that current models do not reliably generalize their reasoning across equivalent instantiations of the same template. This sensitivity to lexical change may also be intensified in a domain like biomedicine. Entities like genes, diseases, and drug names appear at highly heterogeneous frequencies in model pretraining corpora. The effects of this long-tail knowledge phenomenon have been well-studied, showing that LLM performance disproportionately degrades on rare or infrequent entities (Razeghi et al., 2022; Kandpal et al., 2023). As a result, substituting rare biomedical entities within the same template can introduce a meaningful distribution shift and negatively impact performance, even when the logical form of the query remains constant. We also cite the difficulty of the benchmark task despite our use of templates, as top models still trail expert performance by nearly 30%.

**Multiple valid SQL solutions.** While gold SQL queries in BiomedSQL were authored by domain experts and independently verified by analysts, they do not represent the only correct way to retrieve relevant data for a given question. Biomedical questions often permit multiple semantically valid formulations, e.g., using alternative joins, filters, or aggregations. To account for this, we evaluate models using a combination of metrics, including execution-based (EX, JAC) and LLM-judged response quality (RQR), to more robustly reflect real-world answer utility.

**Use of BigQuery SQL.** While we recognize that reliance on a cloud-specific dialect such as BigQuery may limit direct comparability with prior work, we view this as an important design decision. Cloud-native SQL dialects are increasingly common in production systems, especially in biomedical informatics pipelines. Evaluating LLMs in this setting brings new challenges, including vendor-specific functions, syntax, and query planning, that have been underexplored in the research community.

**Future directions.** We plan to expand our set of template questions to cover the experimental 20-table schema. We also have additional tables covering extensive GWAS and CRISPR screen data that would also be a valuable addition to the benchmark. We plan to convert the benchmark to SQLite for easier integration with more general-purpose text to SQL systems such as DIN-SQL (Pourreza and Rafiei, 2023) and CHESS (Talaei et al., 2024). Translating the questions from BiomedSQL into other languages is another avenue for future work that would both increase the difficulty of the benchmark task and increase language equity in the text-to-SQL research community. Finally, a public-facing leaderboard would facilitate researcher attempts to saturate the biomedical reasoning text-to-SQL task.

## 8    CONCLUSION

We present BiomedSQL, the first large-scale text-to-SQL benchmark explicitly designed to evaluate scientific reasoning during SQL generation in the biomedical domain. Our experiments show that BiomedSQL poses a substantial challenge to state-of-the-art LLMs, with execution accuracy and answer quality still lagging far behind domain expert performance.

By focusing on implicit domain conventions, multi-step reasoning, and structured biomedical data, BiomedSQL highlights key limitations of current systems and offers a rigorous testbed for future research. We believe this benchmark is a critical step toward building more capable, trustworthy text-to-SQL systems that can broaden access to biomedical knowledge and accelerate discovery for researchers across disciplines.

## 9    REPRODUCIBILITY STATEMENT

In the Supplementary Material, we have included a compressed folder, biomedsql.zip with our self-contained code to reproduce all of the main experimental results from this paper. This includes service account credentials that provide access to our BigQuery database, the full benchmark dataset of 68,000 question/SQL query/answer triples at `biomedsql/data/BiomedSQL.csv`, and the test set of 546 questions used for the experiments at `biomedsql/data/dev_sample.csv`. In addition, Appendix A.12 details the expected compute resources needed for all experiments.

Upon publication, we plan to push the code to a public GitHub repository, host the benchmark dataset and tabular data that composes our database in a public HuggingFace repository, and make our BigQuery database publicly accessible in order to support reproducible research.

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

# A APPENDIX

## A.1 DATABASE TABLES.

This section provides schema details and a short description of the ten core tables in the BiomedSQL BigQuery database. Database Tables 1-2, 6, and 9-10 are generated from sources that are under the CC0 1.0 License or are otherwise designated to the public domain. Database Tables 7-8 are generated from sources that are under either the CC0 1.0 License (OpenTargets) or the CC Attribution-ShareAlike 3.0 Unported license (ChEMBL). Database Tables 3, 5 and Database Table 4 are generated from sources under the CC BY 4.0 and CC BY-NC 4.0 License, respectively.

Database Table 1: Alzheimer's Disease GWAS (21.1M rows, 13 columns).

```
Table: AlzheimerDisease_CombinedGeneData_UUID
Description: Summary statistics from the largest publicly available GWAS of Alzheimer's Disease in a
European population (Bellenguez et al., 2022).
Schema:
 - Name: UUID | Type: STRING | Mode: REQUIRED
 - Name: SNP | Type: STRING | Mode: NULLABLE
 - Name: A1 | Type: STRING | Mode: NULLABLE
 - Name: A2 | Type: STRING | Mode: NULLABLE
 - Name: freq | Type: FLOAT | Mode: NULLABLE
 - Name: b | Type: FLOAT | Mode: NULLABLE
 - Name: se | Type: FLOAT | Mode: NULLABLE
 - Name: p | Type: FLOAT | Mode: NULLABLE
 - Name: chr_37 | Type: INTEGER | Mode: NULLABLE
 - Name: bp_37 | Type: INTEGER | Mode: NULLABLE
 - Name: chr_38 | Type: INTEGER | Mode: NULLABLE
 - Name: bp_38 | Type: INTEGER | Mode: NULLABLE
 - Name: nearestGene | Type: STRING | Mode: NULLABLE
```

Database Table 2: Parkinson's Disease GWAS (7.8M rows, 13 columns).

```
Table: ParkinsonDisease_CompleteGeneData_No23andMe
Description: Summary statistics from the largest publicly available GWAS of Parkinson's Disease in a
European population (Nalls et al., 2019).
Schema:
 - Name: UUID | Type: STRING | Mode: REQUIRED
 - Name: SNP | Type: STRING | Mode: NULLABLE
 - Name: A1 | Type: STRING | Mode: NULLABLE
 - Name: A2 | Type: STRING | Mode: NULLABLE
 - Name: freq | Type: FLOAT | Mode: NULLABLE
 - Name: b | Type: FLOAT | Mode: NULLABLE
 - Name: se | Type: FLOAT | Mode: NULLABLE
 - Name: p | Type: FLOAT | Mode: NULLABLE
 - Name: chr_37 | Type: INTEGER | Mode: NULLABLE
 - Name: bp_37 | Type: INTEGER | Mode: NULLABLE
 - Name: chr_38 | Type: INTEGER | Mode: NULLABLE
 - Name: bp_38 | Type: INTEGER | Mode: NULLABLE
 - Name: nearestGene | Type: STRING | Mode: NULLABLE
```

Database Table 3: Alzheimer's Disease Gene Pathway Associations (542 rows, 5 columns).

```
Table: AlzheimerDisease_GeneAssoc_Pathways_UUID
Description: Summary statistics from a pathway-level analysis of gene sets in Alzheimer's Disease
(Zhang et al., 2021).
Schema:
 - Name: UUID | Type: STRING | Mode: REQUIRED
 - Name: genes | Type: STRING | Mode: NULLABLE
 - Name: size | Type: INTEGER | Mode: NULLABLE
 - Name: statistic | Type: FLOAT | Mode: NULLABLE
 - Name: p | Type: FLOAT | Mode: NULLABLE
```

Database Table 4: Parkinson's Disease Gene Pathway Associations (1,016 rows, 5 columns).

```
Table: ParkinsonDisease_GeneAssoc_Pathways_UUID
Description: Summary statistics from a pathway-level analysis of gene sets in Parkinson's Disease
(Elango et al., 2023).
Schema:
  - Name: UUID | Type: STRING | Mode: REQUIRED
  - Name: genes | Type: STRING | Mode: NULLABLE
  - Name: size | Type: INTEGER | Mode: NULLABLE
  - Name: statistic | Type: FLOAT | Mode: NULLABLE
  - Name: p | Type: FLOAT | Mode: NULLABLE
```

Database Table 5: Neurodegenerative Disease SMR Associations (1.7M rows, 31 columns).

```
Table: NeurodegenerativeDiseases_SMR_Genes_Full
Description: SMR results providing functional inferences between genetic variants and six neurodegenerative
diseases (Alvarado et al., 2024).
Schema:
  - Name: UUID | Type: STRING | Mode: REQUIRED
  - Name: Omic | Type: STRING | Mode: NULLABLE
  - Name: Disease | Type: STRING | Mode: NULLABLE
  - Name: probeID | Type: STRING | Mode: NULLABLE
  - Name: ProbeChr | Type: INTEGER | Mode: NULLABLE
  - Name: Gene | Type: STRING | Mode: NULLABLE
  - Name: Probe_bp | Type: INTEGER | Mode: NULLABLE
  - Name: topSNP | Type: STRING | Mode: NULLABLE
  - Name: topSNP_chr | Type: INTEGER | Mode: NULLABLE
  - Name: topSNP_bp | Type: INTEGER | Mode: NULLABLE
  - Name: A1 | Type: STRING | Mode: NULLABLE
  - Name: A2 | Type: STRING | Mode: NULLABLE
  - Name: Freq | Type: FLOAT | Mode: NULLABLE
  - Name: b_GWAS | Type: FLOAT | Mode: NULLABLE
  - Name: se_GWAS | Type: FLOAT | Mode: NULLABLE
  - Name: p_GWAS | Type: FLOAT | Mode: NULLABLE
  - Name: b_eQTL | Type: FLOAT | Mode: NULLABLE
  - Name: se_eQTL | Type: FLOAT | Mode: NULLABLE
  - Name: p_eQTL | Type: FLOAT | Mode: NULLABLE
  - Name: b_SMR | Type: FLOAT | Mode: NULLABLE
  - Name: se_SMR | Type: FLOAT | Mode: NULLABLE
  - Name: p_SMR | Type: FLOAT | Mode: NULLABLE
  - Name: p_SMR_multi | Type: FLOAT | Mode: NULLABLE
  - Name: p_HEIDI | Type: FLOAT | Mode: NULLABLE
  - Name: nsnp_HEIDI | Type: FLOAT | Mode: NULLABLE
  - Name: topRSID | Type: STRING | Mode: NULLABLE
  - Name: Omic_type | Type: STRING | Mode: NULLABLE
  - Name: Omic_tissue | Type: STRING | Mode: NULLABLE
  - Name: Disease_name | Type: STRING | Mode: NULLABLE
  - Name: Source | Type: STRING | Mode: NULLABLE
  - Name: func_sig | Type: STRING | Mode: NULLABLE
```

Database Table 6: Neurodegenerative Disease Allele Frequencies (72.2M rows, 6 columns).

```
Table: NeurodegenerativeDisease_AlleleFrequencies_UUID
Description: Allele frequencies from a cohort not containing Alzheimer's or Parkinson's disease cases
(Bergstrom et al., 2020).
Schema:
  - Name: UUID | Type: STRING | Mode: REQUIRED
  - Name: chr_38 | Type: INTEGER | Mode: NULLABLE
  - Name: SNP | Type: STRING | Mode: NULLABLE
  - Name: A1 | Type: STRING | Mode: NULLABLE
  - Name: A2 | Type: STRING | Mode: NULLABLE
  - Name: freq | Type: FLOAT | Mode: NULLABLE
```

Database Table 7: Drug Gene Targets (6,391 rows, 20 columns).

```
Table: DrugGeneTargets_ComprehensiveAnnotations_updated
Description: Details drug-gene relationships and offers a comprehensive view of drug development pipelines
(OpenTargets and ChEMBL).
Schema:
  - Name: UUID | Type: STRING | Mode: REQUIRED
  - Name: chemblIdentifier | Type: STRING | Mode: NULLABLE
  - Name: blackBoxWarning | Type: BOOLEAN | Mode: NULLABLE
  - Name: drugName | Type: STRING | Mode: NULLABLE
  - Name: drugMolecularType | Type: STRING | Mode: NULLABLE
  - Name: yearOfFirstApproval | Type: INTEGER | Mode: NULLABLE
  - Name: maxClinicalTrialPhase | Type: INTEGER | Mode: NULLABLE
  - Name: drugHasBeenWithdrawn | Type: BOOLEAN | Mode: NULLABLE
  - Name: drugIsApproved | Type: BOOLEAN | Mode: NULLABLE
  - Name: tradeNames_string | Type: STRING | Mode: NULLABLE
  - Name: drugSynonyms_string | Type: STRING | Mode: NULLABLE
  - Name: linkedDiseasesDrug_string | Type: STRING | Mode: NULLABLE
  - Name: linkedDiseasesCount | Type: INTEGER | Mode: NULLABLE
  - Name: newLinkedTargets_string | Type: STRING | Mode: NULLABLE
  - Name: numberLinkedTargets | Type: INTEGER | Mode: NULLABLE
  - Name: drugDescription | Type: STRING | Mode: NULLABLE
  - Name: drugActionType | Type: STRING | Mode: NULLABLE
  - Name: drugMechanismOfAction | Type: STRING | Mode: NULLABLE
  - Name: tradename_count | Type: INTEGER | Mode: NULLABLE
  - Name: synonyms_count | Type: INTEGER | Mode: NULLABLE
```

Database Table 8: Drug Target Indications (1.2M rows, 23 columns).

```
Table: DrugTargets_IndicationsAndTherapeuticUses
Description: Links drugs to specific indications, facilitating disease- and target-specific
therapeutic explorations (OpenTargets and ChEMBL).
Schema:
  - Name: UUID | Type: STRING | Mode: REQUIRED
  - Name: chemblId | Type: STRING | Mode: NULLABLE
  - Name: drugName | Type: STRING | Mode: NULLABLE
  - Name: tradeName | Type: STRING | Mode: NULLABLE
  - Name: drugType | Type: STRING | Mode: NULLABLE
  - Name: actionType | Type: STRING | Mode: NULLABLE
  - Name: targetType | Type: STRING | Mode: NULLABLE
  - Name: target | Type: STRING | Mode: NULLABLE
  - Name: approvedSymbol | Type: STRING | Mode: NULLABLE
  - Name: approvedName | Type: STRING | Mode: NULLABLE
  - Name: yearOfFirstApproval | Type: INTEGER | Mode: NULLABLE
  - Name: usan_year | Type: FLOAT | Mode: NULLABLE
  - Name: patent_no | Type: STRING | Mode: NULLABLE
  - Name: max_phase_for_ind | Type: FLOAT | Mode: NULLABLE
  - Name: mesh_id | Type: STRING | Mode: NULLABLE
  - Name: mesh_heading | Type: STRING | Mode: NULLABLE
  - Name: efo_id | Type: STRING | Mode: NULLABLE
  - Name: efo_term | Type: STRING | Mode: NULLABLE
  - Name: tradeNames_list | Type: STRING | Mode: NULLABLE
  - Name: tradename_count | Type: INTEGER | Mode: NULLABLE
  - Name: syns_list | Type: STRING | Mode: NULLABLE
  - Name: synonyms_count | Type: INTEGER | Mode: NULLABLE
  - Name: ct | Type: STRING | Mode: NULLABLE
```

Database Table 9: Drug Licensing (2,097 rows, 16 columns).

```
Table: DrugTargets_LiscensingAndUses
Description: Licensing, pharmaceutical company, and dosage information for specific drugs (FDA Purple Book).
Schema:
  - Name: UUID | Type: STRING | Mode: REQUIRED
  - Name: applicant | Type: STRING | Mode: NULLABLE
  - Name: blaNumber | Type: INTEGER | Mode: NULLABLE
  - Name: tradeName | Type: STRING | Mode: NULLABLE
  - Name: drugName | Type: STRING | Mode: NULLABLE
  - Name: blaType | Type: STRING | Mode: NULLABLE
  - Name: strength | Type: STRING | Mode: NULLABLE
  - Name: dosageForm | Type: STRING | Mode: NULLABLE
  - Name: routeOfAdministration | Type: STRING | Mode: NULLABLE
  - Name: productPresentation | Type: STRING | Mode: NULLABLE
  - Name: marketingStatus | Type: STRING | Mode: NULLABLE
  - Name: licensure | Type: STRING | Mode: NULLABLE
  - Name: submissionType | Type: STRING | Mode: NULLABLE
  - Name: licenseNumber | Type: INTEGER | Mode: NULLABLE
  - Name: productNumber | Type: INTEGER | Mode: NULLABLE
  - Name: center | Type: STRING | Mode: NULLABLE
```

Database Table 10: Drug Dosages (211k rows, 11 columns).

```
Table: DrugTargets_UsesAndDosages
Description: Dosage, route of administration, and strength information for specific drugs
(National Drug Code).
Schema:
  - Name: UUID | Type: STRING | Mode: REQUIRED
  - Name: productType | Type: STRING | Mode: NULLABLE
  - Name: tradeName | Type: STRING | Mode: NULLABLE
  - Name: drugName | Type: STRING | Mode: NULLABLE
  - Name: dosageForm | Type: STRING | Mode: NULLABLE
  - Name: dosageRoute | Type: STRING | Mode: NULLABLE
  - Name: labelerName | Type: STRING | Mode: NULLABLE
  - Name: activeDosage_strength | Type: STRING | Mode: NULLABLE
  - Name: activeIngredient_strength | Type: STRING | Mode: NULLABLE
  - Name: mechanismOfAction_pharma | Type: STRING | Mode: NULLABLE
  - Name: packageDescription | Type: STRING | Mode: NULLABLE
```

## A.2 BIOLOGICAL REASONING CATEGORIES.

Table 6 defines the biological reasoning categories that BiomedSQL challenges and Figure 4 shows their distribution among the set of 68,000 queries. The *GWAS Significance*, *SMR Significance*, and *Functional Significance* categories test the ability of LLMs to operationalize domain-specific statistical significance thresholds. Categories such as *Approval Status*, *Genetic Target*, and *Trial Phase* task LLMs with understanding and applying information about clinical trial phases for specific drugs and indications to generate a correct SQL query.

Table 6: Description of Biological Reasoning Categories in BiomedSQL.

| Bio Category | Description |
|---|---|
| Approval Status | Retrieves information on the FDA approval status of a drug or indication. |
| Trial Phase | Retrieves information on the clinical trial phase a drug has reached. |
| GWAS Significance | Identifies variants that are GWAS significant for a disease ($p < 5e\text{-}08$). |
| SMR Significance | Identifies variants that are SMR significant for a disease ($p < 2.95e\text{-}06$). |
| Functional Significance | Identifies variants that are significant in a particular tissue for a disease. |
| Effect | Retrieves the effect size and direction for specific variants. |
| Genetic Target | Retrieves information on the genetic target of a drug. |
| Allele Frequency | Calculates allele frequencies for a variant or set of variants. |
| Metadata | Retrieves general information on a drug or genetic variant. |

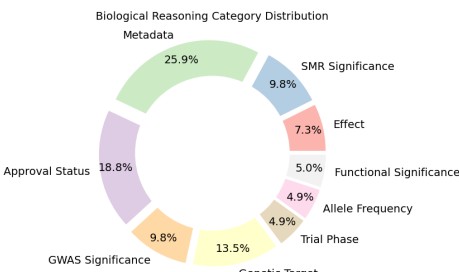

Figure 4: Distribution of biological reasoning query types.

## A.3 ISOLATED SQL GENERATION PROMPT TEMPLATES.

This section provides details about the prompt engineering approaches used for the isolated SQL query generation experiments.

Prompt 1 contains the baseline prompt template that was used. For these experiments, the *db_schema* variable is replaced with the schema that is detailed in Appendix A.1. In the case of the *3-rows*, *5-rows*, and *combo* experiments, *db_schema* is replaced with a schema that includes the corresponding number of example rows for each table in the database.

Prompt 2 shows the example queries that were appended to the baseline prompt for use in the *1-shot*, *3-shot*, *5-shot*, and *combo* experiments. Note that these example queries stay the same regardless of the question from BiomedSQL being passed.

Prompt 3 details the statistical thresholding instructions that were appended to the baseline prompt for use in the *stat-instruct* and *combo* experiments.

Finally, Prompt 4 contains the prompt template that was passed to the LLMs for the generation of a final natural language response based on the question, generated SQL query, and execution results. This prompt was used throughout the isolated SQL generation experiments.

```
You are a data analyst and SQL developer experienced with biomedical data in Google BigQuery.
Your task is to translate the user's natural language question into a syntactically correct
Google BigQuery SQL query.

User's Natural Language Question:
{question}

Database Schema:
{db_schema}

Use these guidelines when generating the query:
    1. Review the database schema.
    2. Review the user's question.
    3. Generate a valid Google BigQuery SQL query that answers the question based on the schema.
    4. Always enclose table references in backticks, e.g. `project.dataset.table`.
    5. Make use of BigQuery-specific functions and syntax where appropriate
    (e.g., DISTINCT, aliases, ORDER BY).
    6. Always include the UUID column in your SELECT statements, except in cases of questions where the
    COUNT and ORDER BY functions are needed.
    7. Unless the user explicitly requests a different LIMIT, default your queries to LIMIT 100.
    8. Output ONLY the raw SQL query (no additional commentary or explanations).
    9. Avoid SELECT *; select only the necessary columns to answer the user's query.
    10. Ensure that any disease names that contain an apostrophe in the query are surrounded by
    double quotes (e.g., "Alzheimer's Disease").

Please only return the SQL query in the following format:
```
{{sql_query}}
```
```

Prompt 1: Baseline prompt template for the isolated SQL generation experiments.

```
Below are example BigQuery queries to guide you (for reference only, do not repeat verbatim unless needed
by the user's request):
Example 1:
    SELECT DISTINCT drugName, drugIsApproved, newLinkedTargets_string
    FROM `card-ai-389220.bio_sql_benchmark.DrugGeneTargets_ComprehensiveAnnotations_updated`
    WHERE newLinkedTargets_string LIKE "%TUBB %"
    AND drugIsApproved = TRUE
    LIMIT 1000;
Example 2:
    SELECT SNP, A1 AS effect_allele, freq AS effect_allele_frequency,
    A2 AS non_effect_allele, 1 - freq AS non_effect_allele_frequency
    FROM `card-ai-389220.bio_sql_benchmark.AlzheimerDisease_CombinedGeneData_UUID`
    WHERE SNP = 'rs61769339'
    LIMIT 10;
Example 3:
    SELECT drugName, newLinkedTargets_string, drugIsApproved
    FROM `card-ai-389220.bio_sql_benchmark.DrugGeneTargets_ComprehensiveAnnotations_updated`
    WHERE newLinkedTargets_string LIKE '%ACACA%'
    AND drugIsApproved = TRUE
    LIMIT 1000;
Example 4:
    SELECT topRSID, Disease, Gene, p_SMR_multi, p_HEIDI, b_SMR
    FROM `card-ai-389220.bio_sql_benchmark.NeurodegenerativeDiseases_SMR_Genes_Full`
    WHERE Disease = 'FTD' AND Gene = 'ORC3' AND p_SMR_multi < 2.95e-6
    LIMIT 100
Example 5:
    SELECT DISTINCT drugName, tradeNames_list, drugType, actionType, target, approvedSymbol, approvedName,
    yearOfFirstApproval, max_phase_for_ind, mesh_heading, efo_term
    FROM `card-ai-389220.bio_sql_benchmark.DrugTargets_IndicationsAndTherapeuticUses`
    WHERE (LOWER(efo_term) = "acute hepatic porphyria"
    OR LOWER(mesh_heading) = "acute hepatic porphyria") AND LOWER(drugType) = "oligonucleotide"
    AND yearOfFirstApproval > 0 AND max_phase_for_ind = 4.0
    LIMIT 100
```

Prompt 2: Example queries appended to the baseline prompt for the *n-shot* and *combo* experiments.

```
Use the following p-value thresholds for questions about statistical significance:
    1. p < 5e-8 for genome-wide significance.
    2. p_SMR < 2.95e-6 for SMR significance.
    3. p_SMR < 2.95e-6, p_HEIDI > 0.01 for functional significance.
```

Prompt 3: Statistical thresholding instructions appended to the baseline prompt for the *stat-instruct* and *combo* experiments.

```
You are a data analyst and SQL developer experienced with biomedical data in Google BigQuery.
Given the following question, SQL query, and SQL query execution results, please provide a concise answer.
Please do not use any information outside of the SQL query and SQL query execution results in your answer.
Question:
{question}
SQL Query:
{sql_query}
Execution Results:
{execution_results}
```

Prompt 4: Natural language response prompt template for the isolated SQL generation experiments.

## A.4    REACT PROMPT TEMPLATE.

Prompt 5 shows the ReAct-style prompt template used in the interaction paradigm experiments. Similar to the baseline prompt *db_schema* is replaced with the schema that is detailed in Appendix A.1. *history_str* is replaced by the reasoning trace from previous steps that the LLM chooses to take. We allow the LLM to perform up to 5 iterations within the ReAct loop before a final answer is generated.

```
You are an expert SQL agent that uses step-by-step reasoning to answer questions about data in a
BigQuery database.

IMPORTANT: The dataset name is "{dataset_name}". Always qualify table names with this dataset name.
Example: SELECT * FROM {dataset_name}.table_name

Follow these steps:
    1. Think about how to translate the question into a SQL query.
    2. Decide which tables and columns are needed.
    3. Write a SQL query with explanatory comments.
    4. Verify the query syntax before executing.
    5. If the query has errors, fix them and try again.
    6. Once the query is successful, explain the results clearly.
    7. Always include the UUID column in your SELECT statements, except in cases of questions where
    the COUNT and ORDER BY functions are needed.
    8. Unless the user explicitly requests a different LIMIT, default your queries to LIMIT 100.
    9. Avoid SELECT *; select only the necessary columns to answer the user's query.
    10. Ensure that any disease names that contain an apostrophe in the query are surrounded by
    double quotes (e.g., "Alzheimer's Disease").

Your output MUST be a JSON object with these fields:
{{
    "thought": "Your reasoning about how to answer the question",
    "action": "One of 'verify_sql', 'execute_sql', or 'final_answer'",
    "action_input": "For verify_sql/execute_sql: the SQL query;
                     For final_answer: explanation of the results"
}}

IMPORTANT:
    - Your response must include valid JSON that can be parsed.
    - Do not include any explanations outside the JSON object.
    - Always qualify table names with the dataset name "{dataset_name}."

Make sure your SQL queries follow BigQuery SQL syntax and include helpful inline comments.

Question: {question}

Database Schema:
```
{db_schema}
```

Reasoning History:
{history_str}

Continue the reasoning process with the next step:
```

Prompt 5: ReAct prompt template for the interaction paradigm experiments.

## A.5 BMSQL PROMPT TEMPLATES.

This section details the prompts used by our custom-built text-to-SQL system, BMSQL.

Prompt 6 provides the template for the first step in the BMSQL pipeline, which is using the schema to identify relevant tables and columns to generate a SQL query given the question. Once relevant columns are selected, Prompt 7 is used for BMSQL to generate a first attempt at a general SQL query that corresponds to the question. If the execution of this query fails, Prompt 8 is used to generate a query that resolves any syntax errors present in the original query. BMSQL is given up to three retries to correct any syntax errors at this step.

Prompt 9 is used to generate a query that applies any statistical thresholding rules that may be necessary to answer the question. If no statistical thresholding is needed, the general query is returned once again. Using the execution results from both the general and refined query, BMSQL is asked to generate a final response to the question given the instructions in Prompt 10.

Finally, Prompt 11 is used in the inference time compute experiments to give BMSQL an opportunity to deem the final response as insufficient to answer the question and take subsequent passes through the pipeline. On these subsequent passes, BMSQL tends to correct any syntax errors but rarely makes structural changes to the generated SQL queries.

The multi-stage query generation that BMSQL uses was designed to reflect how a domain expert might query a biomedical knowledge base by first checking if data is available for a given query, and applying statistical thresholding on a subsequent query. As seen throughout §5.2, BMSQL is a top performer in terms of both execution metrics and response quality.

```
You are a BioMedical Domain Expert with deep database knowledge. You have the following database schema:
{db_schema}

The user has asked a question about this biomedical data:
"{question}"

Your task:
    1. Identify the single table or multiple tables (if absolutely necessary) that would provide the
    *full* answer to this question.
    2. From these table(s), list *all columns* that might be relevant to fully answer the question.
    (Because a downstream aggregator will handle details, do NOT omit columns that may be relevant.)

Format your response **strictly** as:
TABLE_NAME: col1, col2, col3, ...
    - Provide no extra commentary or text.
    - If multiple tables are truly needed, list each in a new line, in the same format.
```

Prompt 6: BMSQL prompt template for selecting relevant columns.

```
You are a highly proficient BigQuery SQL generator in the biomedical domain.

Database schema:
{db_schema}

The user asked:
"{question}"

Previously identified relevant columns/tables:{relevant_columns}

Instructions:
    - Generate exactly one valid BigQuery SQL query that retrieves all relevant columns
    from the relevant_columns list.
    - Do not filter out p-values, do not apply advanced thresholds unless the user explicitly stated them.
    - If the user mentions FDA approval, include those columns.
    - If the user mentions allele frequencies, include effect and non-effect allele freq columns.
    - FROM clause: `{project_id}.{dataset_name}.table_name`
    - Always include the UUID column in your SELECT statements, except in cases of questions where
    the COUNT and ORDER BY functions are needed.
    - Unless the user explicitly requests a different LIMIT, default your queries to LIMIT 100.

Return only the final SQL in a markdown code block:
```sql
{{sql_query}}
```
```

Prompt 7: BMSQL prompt template for generating a first attempt general SQL query.

```
You are a SQL debugging assistant for Google BigQuery. Below is the database schema,
the failed query, and the error message or unexpected results:

=== DATABASE SCHEMA START ===
{db_schema}
=== DATABASE SCHEMA END ===

=== FAILED SQL QUERY START ===
```sql
{general_query}
```
=== FAILED SQL QUERY END ===

=== ERROR OR RESULTS START ===
{general_results}
=== ERROR OR RESULTS END ===

The user originally asked:
"{question}"

Relevant columns identified for answering this question:
{relevant_columns}

Your task:
    - Analyze the failed query and the error or result details.
    - Generate a corrected SQL query that resolves the issue,
    ensuring it's correct for BigQuery and fits the schema.

Format the corrected query as a valid SQL query in a markdown fenced block:
```sql
{{sql_query}}
```
```

Prompt 8: BMSQL prompt template for correcting a failed first attempt general SQL query.

```
You are a skillful BigQuery SQL refiner. The user might want additional thresholds or
see if there's advanced filtering needed, e.g. p-values or FDA approvals.

Original question: "{question}"

The previously generated SQL query was:
```sql
{sql_query}
```

The query's results (showing up to 10 rows):
{resp_str}

Database schema:
{db_schema}

Known threshold rules:
{threshold_rules}

If no extra thresholds or filters are implied, keep the same query.
Otherwise, produce a refined SQL with the new filters,
returning it in a markdown code block:
```sql
{{sql_query}}
```
```

Prompt 9: BMSQL prompt template for generating a refined SQL query that applies thresholding rules if necessary.

```
You are a BioMedical Domain expert that is returning a concise answer to the user's question
based on two sets of SQL queries and results. If not sure, say you do not know.

Question: {question}

SQL query 1: {sql_query_1}
Result 1: {result_1}

SQL query 2: {sql_query_2}
Result 2: {result_2}
```

Prompt 10: BMSQL prompt template for generating a natural language response to the question.

```
You are a biomedical domain and BigQuery expert that is determining if a text-to-SQL workflow
should be run again.
Based on the question SQL queries, their execution results, and the final answer,
determine if you are confident in the answer.

Use the following guidelines:
    1. If the SQL queries or answer contain errors, deem the answer as insufficient.
    2. If you have any doubts about the SQL queries, execution results, or answer,
    deem the answer as insufficient.
    3. If there are any inconsistencies between the SQL queries, execution results, and answer,
    deem the answer as insufficient.
    4. Keep in mind that a negative answer (i.e. "No, ...") does not necessarily mean
    the answer is insufficient.
    5. Otherwise, use your best judgement.
    6. Do not use any external information outside of what is provided.

Question: {question}

SQL query 1: {sql_query_1}
Result 1: {result_1}

SQL query 2: {sql_query_2}
Result 2: {result_2}

Answer: {answer}

Please only return 'Yes' if the answer is sufficient and 'No' if it is insufficient.
```

Prompt 11: BMSQL prompt template for determining if an answer is sufficient and taking a subsequent pass at the pipeline if not.

## A.6 EVALUATION METRIC DEFINITIONS.

We provide the formulaic definitions for the evaluation metrics described in §5.1.

**Execution Accuracy (EX).** Given two sets of SQL execution results, the reference set $R_n$ produced by the $n$ ground-truth queries, and the corresponding result set $\hat{R}_n$ produced by the $n$ LLM-generated queries, EX can be computed as follows:

$$EX = \frac{\sum_{n=1}^{N} \mathbb{I}(r_n, \hat{r}_n)}{N} \quad (1)$$

$$\text{where} \quad \mathbb{I}(r_n, \hat{r}_n) = \begin{cases} 1, & \text{if } r_n = \hat{r}_n \\ 0, & \text{otherwise} \end{cases} \quad \text{and} \quad r_n \in R_n, \hat{r}_n \in \hat{R}_n \quad (2)$$

**Jaccard Index (JAC).** Given two sets of SQL execution results, the reference set $R_n$ produced by the $n$ ground-truth queries, and the corresponding result set $\hat{R}_n$ produced by the $n$ LLM-generated queries, JAC can be computed as follows:

$$JAC = \frac{\sum_{n=1}^{N} \mathbb{J}(r_n, \hat{r}_n)}{N} \quad (3)$$

$$\text{where} \quad \mathbb{J}(r_n, \hat{r}_n) = \frac{|r_n \cap \hat{r}_n|}{|r_n \cup \hat{r}_n|} \quad \text{and} \quad r_n \in R_n, \hat{r}_n \in \hat{R}_n \quad (4)$$

**Syntax Error Rate (SER).** Given a set of LLM-generated SQL queries $\hat{R}_n$ resulting from $n$ questions in BiomedSQL, SER can be computed as follows:

$$SER = \frac{\sum_{n=1}^{N} \mathbb{E}(\hat{r}_n)}{N} \quad (5)$$

$$\text{where} \quad \mathbb{E}(\hat{r}_n) = \begin{cases} 1, & \text{if } exec(\hat{r}_n) = Error \\ 0, & \text{otherwise} \end{cases} \quad \text{and} \quad \hat{r}_n \in \hat{R}_n \quad (6)$$

$$\text{and} \quad exec(\hat{r}_n) \quad \text{is the result of running the generated SQL query on the database.} \quad (7)$$

**BioScore.** Prompt 12 contains the prompt template for generating BioScore, the LLM-as-a-judge metric used to grade the quality of a natural language response compared to the gold standard response. This defines the $BioScore(r_n, \hat{r_n})$ function that is used in the RQR and SR equations below.

```
You are a highly knowledgeable and experienced expert in the healthcare and biomedical field, possessing
extensive medical knowledge and practical expertise.

Scoring Instructions for Evaluating Analyst Responses

Objective: Evaluate an analyst's response against a gold standard.

Scoring Criteria:
- Exact Match: 3 points for an exact or equally accurate response.
- Close Match: 2 points for a very close response with minor inaccuracies.
- Partial Match: 1 point for a partially accurate response with significant omissions.
- Irrelevant Information (Harmless): Deduct 0.5 points for harmless irrelevant information.
- Irrelevant Information (Distracting): Deduct 1 point for distracting irrelevant information.
- No Match: 0 points for no match.
- Not Knowing Response: -1 point for stating lack of knowledge or abstaining.
  An example of this scenario is
  when Analyst Response says 'There are various studies, resources or databases on this topic that you can
  check ... but I do not have enough information on this topic.'

Scoring Process:
    1. Maximum Score: 3 points per question.
    2. Calculate Score: Apply criteria to evaluate the response.

Question: {question}
Golden Answer: {gold_ans}
Analyst Response: {pred_ans}

Your grading
Using the scoring instructions above, grade the Analyst Response.
Return only the numeric score on a scale from 0.0-3.0.
If the response is stating lack of knowledge or abstaining, give it -1.0.
Please respond only with the score.
```

Prompt 12: BioScore prompt template.

**Response Quality Rate (RQR).** Given two sets of natural language responses, the reference set $R_n$ which map to $n$ questions in BiomedSQL, and the corresponding result set $\hat{R}_n$ containing $n$ LLM-generated responses, RQR can be computed as follows:

$$RQR = \frac{\sum_{n=1}^{N} Quality(r_n, \hat{r}_n)}{N} \tag{8}$$

$$\text{where} \quad Quality(r_n, \hat{r}_n) = \begin{cases} 1, & \text{if} \quad BioScore(r_n, \hat{r}_n) \geq 2 \\ 0, & \text{otherwise} \end{cases} \quad \text{and} \quad r_n \in R_n, \hat{r}_n \in \hat{R}_n \tag{9}$$

**Safety Rate (SR).** Given two sets of natural language responses, the reference set $R_n$ which map to $n$ questions in BiomedSQL, and the corresponding result $\hat{R}$ containing $n$ LLM-generated responses, SR can be computed as follows:

$$SR = \frac{\sum_{n=1}^{N} \mathbb{A}(r_n, \hat{r}_n)}{\sum_{n=1}^{N} \mathbb{B}(r_n, \hat{r}_n)} \tag{10}$$

$$\text{where} \quad \mathbb{A}(r_n, \hat{r}_n) = \begin{cases} 1, & \text{if} \quad BioScore(r_n, \hat{r}_n) = -1 \\ 0, & \text{otherwise} \end{cases} \tag{11}$$

$$\text{and} \quad \mathbb{B}(r_n, \hat{r}_n) = \begin{cases} 1, & \text{if} \quad Bioscore(r_n, \hat{r}_n) < 2 \\ 0, & \text{otherwise} \end{cases} \quad \text{and} \quad r_n \in R_n, \hat{r}_n \in \hat{R}_n \tag{12}$$

## A.7 CORRELATION BETWEEN SQL EXECUTION METRICS AND BIOSCORE METRICS.

In order to further motivate the use of execution-based and BioScore-based response quality metrics, we present a discussion of the correlation between the two. Figure 5 shows heatmaps of both EX (left) and binned JAC (right) compared to our LLM-as-a-judge metric BioScore for the baseline experiment using GPT-o3-mini. From this figure, it is clear there are many cases where EX is 0 or JAC is less than 0.5 and a perfect BioScore is still achieved. This can happen for a variety of reasons, including the presence of negative answers within BiomedSQL (i.e. associations with no significant variants or drug targets without approval), questions where a superset of the correct rows can be returned with a partially correct SQL query (i.e. using a correct ORDER BY clause without applying the correct thresholding values), and cases where there are multiple valid SQL solutions (as disucssed in §7). In these cases the LLM may generate a query that scores poorly in terms of execution metrics but scores adequately in terms of response quality. To mitigate concern for these cases and quantify the association between the execution metrics and BioScore we perform a Cramer's V test which reveals a moderate-to-strong, statistically significant association for both EX (V=0.48, p=$1.84\mathrm{e}-25$) and JAC (V=0.37, p=$3.23\mathrm{e}-39$). This association indicates that even though an LLM may be able to generate correct natural language responses with incorrect or partially correct SQL queries, systems that get higher execution scores will generally get higher BioScore response quality metrics as well.

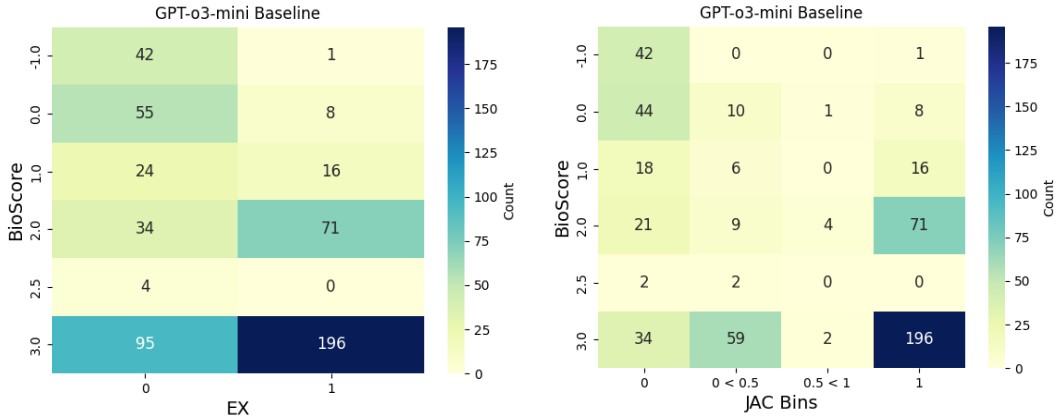

Figure 5: Heatmaps visualizing the association between (left) EX and BioScore and (right) JAC and BioScore.

## A.8 CORRELATION BETWEEN LLM-GENERATED AND DOMAIN-EXPERT GENERATED BIOSCORES.

Since there may be some concern over LLM-as-a-judge metrics yielding unstable assessment results, we present a further justification for our use of BioScore. The prompt for BioScore detailed in Appendix A.6 was generated from a rubric that was used by a domain expert for preliminary evaluations of the natural language responses to questions in BiomedSQL from the LLMs. To demonstrate this association, we took a sample of 100 LLM-generated natural language responses from the experiments throughout the paper. Sampling answers from a variety of different models and interaction paradigms allows us to capture a wide range of failure modes presented, increasing our confidence in the generalization of this analysis across the thousands of questions tested in our experiments. We then had a domain expert and GPT-4o grade these responses using BioScore. We compare the counts of their respective scores in Table 7. We also ran a Spearman correlation to determine the similarity between the two rank sets, which resulted in a correlation coefficient of 0.89 (p < 1e-5). This high level of correlation between the domain expert and LLM-generate BioScores gives a high level of confidence that the LLM-decision based metrics used throughout the paper are both stable and accurate.

Table 7: Comparison of domain expert and GPT-4o BioScores on 40 randomly sampled questions.

| BioScore | Domain Expert | GPT-4o |
|---|---|---|
| -1 | 22 | 21 |
| 0 | 15 | 13 |
| 1 | 8 | 9 |
| 1.5 | 0 | 4 |
| 2 | 21 | 18 |
| 2.5 | 1 | 5 |
| 3 | 33 | 30 |

## A.9 PROMPT EXPERIMENTS.

Table 8 shows the results of the top performing baseline model, GPT-o3-mini, across the range of prompting experiments described in §5.1. As discussed in §5.2, *10-shot is the prompting experiment that provides the most substantial gains* in terms of both execution metrics and response quality compared to the baseline.

Table 8: Performance of GPT-o3-mini in an isolated SQL generation setting using additional prompts.

| Model | EX (%) ↑ | JAC (%) ↑ | RQR (%) ↑ | SR (%) ↑ | SER (%) ↓ | # Tokens |
|---|---|---|---|---|---|---|
| 3-rows | 54.0 (±4.2) | 61.8 (±3.8) | 75.1 (±3.6) | 16.9 (±3.1) | 0.0 (±0.0) | 10,951 |
| 5-rows | 54.2 (±4.2) | 61.9 (±3.8) | 76.7 (±3.5) | 15.7 (±3.0) | 0.2 (±0.4) | 14,312 |
| 1-shot | 54.0 (±4.2) | 61.7 (±3.8) | 73.4 (±3.7) | 32.4 (±3.9) | 0.4 (±0.5) | 4,058 |
| 3-shot | 56.0 (±4.2) | 64.2 (±3.7) | 74.0 (±3.7) | **33.1 (±3.9)** | 0.0 (±0.0) | 4,099 |
| 5-shot | 57.9 (±4.1) | 65.6 (±3.7) | 75.8 (±3.6) | 23.5 (±3.5) | 0.0 (±0.0) | 4,566 |
| 10-shot | **61.3 (±4.1)** | **67.7 (±3.7)** | **80.3 (±3.3)** | 15.7 (±3.0) | 0.2 (±0.4) | 5,430 |
| 20-shot | 60.5 (±4.0) | 66.6 (±3.7) | 76.8 (±3.5) | 10.7 (±2.6) | 0.0 (±0.0) | 6,535 |
| 40-shot | 61.0 (±4.1) | 67.5 (±3.7) | 78.8 (±3.3) | 18.1 (±3.2) | 0.0 (±0.0) | 8,592 |
| stat-instruct | 57.5 (±4.1) | 64.2 (±3.7) | 73.4 (±3.7) | 31.7 (±3.9) | **0.0 (±0.0)** | 3,456 |
| combo | 59.0 (±4.1) | 66.1 (±3.7) | 77.8 (±3.5) | 24.0 (±3.6) | 0.2 (±0.4) | 10,284 |

## A.10 COMMON SQL ERRORS AMONG TOP-PERFORMING MODELS.

We provide more precise definitions of the SQL error categories introduced in §6:

- **Incorrect Tables:** The generated SQL query used the incorrect table, performed an unnecessary join of tables, or performed an incorrect join of tables.
- **Missing Threshold:** The generated SQL query was missing a significance, clinical trial phase, or other threshold.
- **Incorrect Threshold:** The generated SQL query used an incorrect significance, clinical trial phase, or other threshold.
- **Incorrect Aggregations:** The generated SQL query did not use the necessary aggregations or used the necessary aggregations incorrectly.
- **Syntax Error:** The generated SQL query was syntactically or schematically incorrect and could not be run on the database.

Table 9 shows the distribution of errors made by six of the top-performing models from our experiments. As discussed in §6, incorrect table selection and the improper application of statistical thresholds were the most common errors committed by the LLMs.

## A.11 LARGER SCHEMA RESULTS.

Table 10 shows the results of running the GPT-o3-mini powered models on a larger, 20-table schema as described in §6. As expected, as more tables are introduced, model performance decreases.

Table 9: SQL error category analysis for six of the top-performing models.

| Model | Incorrect Tables | Missing Threshold | Incorrect Threshold | Incorrect Aggregations | Syntax Error | Total |
|---|---|---|---|---|---|---|
| Baseline-GPT-4o | 131 | 63 | 34 | 16 | 7 | 251 |
| Baseline-GPT-o3-mini | 114 | 61 | 36 | 17 | 1 | 229 |
| Baseline-Claude-3.7-sonnet | 192 | 61 | 36 | **0** | 9 | 298 |
| Combo-GPT-o3-mini | 121 | 29 | **4** | 14 | 1 | **169** |
| ReAct-GPT-o3-mini | **99** | 61 | 36 | 11 | **0** | 207 |
| BMSQL-GPT-o3-mini | 118 | **23** | 17 | 8 | 14 | 180 |

However, it is important to note that our custom-built system BMSQL is more robust to the larger schema than the single-turn, prompt-based approaches tested.

Table 10: Results for models run on larger schema.

| Model | EX (%) ↑ | JAC (%) ↑ | RQR (%) ↑ | SR (%) ↑ | SER (%) ↓ | # Tokens |
|---|---|---|---|---|---|---|
| Baseline-GPT-o3-mini | 45.8 (±4.2) | 50.7 (±4.0) | 64.1 (±4.0) | 37.8 (±4.1) | 0.4 (±0.5) | 6,201 |
| Combo-GPT-o3-mini | 51.1 (±4.2) | 56.3 (±4.0) | 69.0 (±3.9) | **45.6 (±4.2)** | 0.5 (±0.6) | 19,994 |
| BMSQL-GPT-o3-mini | **58.4 (±4.1)** | **65.1 (±3.7)** | **81.0 (±3.3)** | 37.5 (±4.1) | 1.1 (±0.9) | 108,145 |

## A.12 DECLARATION OF LLM USAGE.

LLMs were used to assist in the preparation of this manuscript. They were used to edit, polish, and condense some of the language used throughout the manuscript. Additionally, LLMs were used to edit code to create some of the figures that appear in the manuscript. The authors take full responsibility for the contents of this work.

## A.13 EXPECTED COMPUTE RESOURCES.

Table 11 details the compute resources needed to reproduce all of the described experiments. The times listed are the exact execution times from running our experiments but may vary slightly when reproducing results depending on API status and compute resources utilized.

Table 11: Compute resources needed to reproduce all experiments. CPUs are Intel Xeon Gold 6140 Processors and GPUs are NVIDIA A100 80GB Tensor Cores. Times are listed in terms of Hours:Minutes. Costs are listed in terms of USD. *Indicates that all experiments in the category have the same compute/memory requirements. NA indicates the experiment was run on compute cluster GPUs and costs could not be estimated.

| Experiment | Model | Compute | Memory | Time | Cost |
|---|---|---|---|---|---|
| **Baseline** | GPT-4o | 2 CPUs | 16GB RAM | 0:44 | $5.03 |
| | GPT-4o-mini | 2 CPUs | 16GB RAM | 1:35 | $7.34 |
| | GPT-o3-mini | 2 CPUs | 16GB RAM | 4:38 | $2.37 |
| | Gemini-2.0-flash | 2 CPUs | 16GB RAM | 0:40 | $0.20 |
| | Gemini-2.0-flash-lite | 2 CPUs | 16GB RAM | 0:37 | $0.13 |
| | Claude-3.7-sonnet | 2 CPUs | 16GB RAM | 1:21 | $6.23 |
| | Qwen-2.5-Coder-14B | 2 GPUs | 32GB VRAM | 1:40 | NA |
| | Qwen-2.5-Coder-32B | 2 GPUs | 64GB VRAM | 2:29 | NA |
| | Llama-3.1-70B | 3 GPUs | 140GB VRAM | 3:17 | NA |
| | Llama-3.1-405B | 2 CPUs | 16GB RAM | 3:21 | $10.05 |
| **Prompt Variations** | 3-rows | 2 CPUs* | 16GB RAM* | 4:18 | $6.58 |
| | 5-rows | | | 4:07 | $8.60 |
| | 1-shot | | | 4:16 | $2.43 |
| | 3-shot | | | 4:09 | $2.46 |
| | 5-shot | | | 6:30 | $2.74 |
| | 10-shot | | | 4:38 | $3.26 |
| | 20-shot | | | 4:54 | $3.92 |
| | 40-shot | | | 5:02 | $5.16 |
| | stat-instruct | | | 2:57 | $2.07 |
| | combo | | | 2:38 | $6.18 |
| **Interaction Paradigms** | ReAct-GPT-4o | 2 CPUs* | 16GB RAM* | 2:01 | $19.50 |
| | ReAct-GPT-o3-mini | | | 4:25 | $8.00 |
| | ReAct-Gemini | | | 2:10 | $0.72 |
| | Index-GPT-4o | | | 1:13 | $1.51 |
| | Index-GPT-o3-mini | | | 3:22 | $1.14 |
| | Index-Gemini | | | 1:11 | $0.04 |
| | DAIL-SQL-GPT-4o | | | 1:38 | $2.22 |
| | DAIL-SQL-GPT-o3-mini | | | 2:55 | $0.72 |
| | DAIL-SQL-Gemini | | | 1:27 | $0.07 |
| | BMSQL-GPT-4o | | | 1:44 | $44.80 |
| | BMSQL-GPT-o3-mini | | | 5:17 | $23.71 |
| | BMSQL-Gemini | | | 1:05 | $1.20 |
| **Inference-time Compute** | 1-pass | 2 CPUs* | 16GB RAM* | 5:17 | $23.71 |
| | 2-pass | | | 6:27 | $32.30 |
| | 3-pass | | | 6:33 | $33.60 |

