# OpenReview forum: "BiomedSQL: Text-to-SQL for Scientific Reasoning on Biomedical Knowledge Bases"
_ICLR.cc/2026/Conference — ICLR 2026 Conference Desk Rejected Submission_

### Official Review · Reviewer_sdka · 2025-10-23

**Soundness:** 3
**Presentation:** 4
**Contribution:** 3
**Rating:** 6
**Confidence:** 4

**Summary:**

The paper “BiomedSQL: Text-to-SQL for scientific reasoning on biomedical knowledge bases” presents a large-scale benchmark aimed at evaluating scientific reasoning in text-to-SQL systems. The authors harmonize several major biomedical knowledge bases (OpenTargets, ChEMBL, GWAS Catalog, omicSynth) into a unified BigQuery schema (~10 core tables), then create roughly 68 000 natural-language question / gold SQL / answer triples derived from 40 expert templates.

The benchmark is used to evaluate numerous LLMs (open + closed models) across single-turn and multi-step paradigms (ReAct, Schema Indexing, DAIL-SQL, and their own BMSQL agent). The work also introduces BioScore, an LLM-based judge for assessing natural-language answers when multiple SQL solutions can be correct.

Overall, BiomedSQL fills an important gap at the intersection of biomedical NLP and structured reasoning.

**Strengths:**

## Strengths

- **Novel and impactful problem scope**
  Text-to-SQL has been widely studied, but very few benchmarks target scientific reasoning or biomedical domains where implicit conventions (e.g., significance thresholds, trial phases) matter. This benchmark addresses that gap.

- **Realistic, large-scale data integration**
  The authors construct a multi-table biomedical schema from authentic public datasets, which lends realism far beyond toy databases used in Spider-style corpora.

- **Extensive empirical coverage**
  Evaluation spans many modern LLMs and multiple reasoning paradigms. Metrics include execution accuracy (EX), Jaccard similarity (JAC), schema-error rate (SER), and BioScore, giving a well-rounded picture.

- **Clear technical writing and reproducibility reporting**
  The paper gives schema diagrams, query examples, and compute details. Appendix sections provide enough detail for independent reproduction.

- **Valuable analysis**
  The authors break down performance by reasoning category (aggregation, multi-table joins, scientific thresholds, etc.) and analyze failure patterns.

**Weaknesses:**

1. **Security / artifact hygiene**
   The supplementary materials appear to contain active service-account credentials for BigQuery access. Even if these are limited in scope, publishing any cloud credentials is a critical security issue.

2. **Licensing and redistribution clarity**
   Several component datasets (e.g., ChEMBL) carry CC BY-NC 4.0 licenses that restrict commercial redistribution. The paper should explicitly list the license for each source table and explain what will be hosted directly versus reconstructed locally.

3. **Template-based generation and linguistic diversity**
   All 68,000 questions are created from 40 expert templates. That’s acceptable for structured benchmarking, but the paper should quantify linguistic variability and discuss how well templated language covers real user queries. If possible, evaluate a small human-written set to test generalization.

4. **LLM-as-judge (BioScore) transparency**
   BioScore is interesting but somewhat underspecified. The paper briefly mentions a small human correlation study; sample size and agreement statistics should be provided to establish reliability. Even basic numbers (n, ρ/κ) would help readers assess the trustworthiness of this metric.

5. **Accessibility / BigQuery dependence**
   Currently all queries execute in BigQuery. For full reproducibility, a local version (e.g., SQLite or CSV dump) should be released or an automated script provided. Otherwise replication requires commercial cloud access.

**Questions:**

1. Can you confirm whether the BigQuery credentials in the supplement are active or dummy placeholders?

2. Please provide a clear per-table license map indicating which datasets are redistributed vs. reconstructed.

3. Will you release a local/offline version (e.g., SQLite or CSV) or reproducible build scripts for the database?

4. Could you elaborate on BioScore reliability — sample size, human–LLM agreement, and any bias checks?

5. How diverse are the 40 question templates linguistically, and do you have results on any human-written queries?

6. Please confirm that all data are public, aggregated, and contain no personally identifiable information.

---

> ### Author Response · Authors · 2025-11-17
>
> Thank you for your time and effort in reviewing our paper. We appreciate your praise of the “novel and impactful problem scope”, “extensive empirical coverage”, and “valuable analysis” of failure patterns that our work provides.
>
> The key contributions of our paper are as follows:
>
> - Benchmark: We introduce BiomedSQL, a benchmark of 68,000 question-SQL query-answer triples designed to evaluate scientific reasoning capabilities of text-to-SQL systems on a realistic, multi-table biomedical database.
> - Infrastructure: We will publish the curated tables that compose our database, expert-written gold queries, execution results, and a toolkit for evaluation and agent orchestration.
> - Evaluation: We benchmark a range of models and interaction paradigms, including a custom-built text-to-SQL system (BMSQL), revealing a 30-40% gap compared to domain expert-level performance.
>
> Below are the responses to your comments:
>
> > Can you confirm whether the BigQuery credentials in the supplement are active or dummy placeholders?
> >
>
> > Will you release a local/offline version (e.g., SQLite or CSV) or reproducible build scripts for the database?
> >
>
> The BigQuery credentials are activate and provide read-only access to the database backing our benchmark, allowing reviewers to reproduce our evaluations. Upon publication, we will revoke this service account and instead release code to recreate the BigQuery database from the original tables.
>
> We plan to release an offline SQLite version of our database and discuss this in Future directions (S7).
>
> > Several component datasets (e.g., ChEMBL) carry CC BY-NC 4.0 licenses that restrict commercial redistribution.
> >
>
> On their website, ChEMBL is currently under the highly permissive CC Attribution-ShareAlike 3.0 Unported license. We also want to note that BiomedSQL is ***not*** intended for commercialization and therefore we do not see the CC BY-NC 4.0 license used by one of our data sources to be a hinderance.
>
> > Please provide a clear per-table license map indicating which datasets are redistributed vs. reconstructed.
> >
>
> In Appendix A.1, we list the license associated with each of the database tables used in our work.
>
> > Could you elaborate on BioScore reliability — sample size, human–LLM agreement, and any bias checks?
> >
>
> The 40 responses used for the correlation analysis in the paper were sampled from multiple models and interaction settings to capture diverse failure modes. This heterogeneity increases confidence that the observed agreement generalizes across thousands of queries with varying model behaviors.
>
> To increase confidence in this agreement, we have expanded the set of domain expert-annotated questions to 100. A table containing the new comparison counts is presented below:
>
> | BioScore | Domain Expert | GPT-4o |
> | --- | --- | --- |
> | -1 | 22 | 21 |
> | 0 | 15 | 13 |
> | 1 | 8 | 9 |
> | 1.5 | 0 | 4 |
> | 2 | 21 | 18 |
> | 2.5 | 1 | 5 |
> | 3 | 33 | 30 |
>
> These new counts resulted in a Spearman correlation of 0.89 (p <1e-5). This expanded correlation analysis further increases our confidence that the high agreement between LLM-generated and expert-generated BioScores will hold across thousands of queries.
>
> We added a description of this experiment in the main text (S5.1) and in A.8 to ensure it is prevalent to the reader. We also inserted the table above in place of Table 7. In the newly attached PDF, all additions to the text are in blue and all redactions are in red to ensure it is clear where changes were made.
>
> > How diverse are the 40 question templates linguistically, and do you have results on any human-written queries?
> >
>
> To quantify the linguistic diversity of our templates, we computed cross-template Jaccard similarity. This measures the fraction of shared vocabulary between two template questions (i.e. the size of the intersection divided by the size of the union of their word sets). Across the 40 template questions, we observed a median unigram similar of 0.08, and no shared bigrams. This implies that the templates share only about 8% of their words and no recurring two-word phrases, demonstrating that the templates are lexically and phrasally distinct.
>
> Creating and validating a new human-authored question set requires biologically meaningful task design and careful SQL verification. Producing a substantial new set takes weeks of expert effort and is beyond our current scope. We outline plans to expand to a 20-table schema in Section 7.
>
> > Please confirm that all data are public, aggregated, and contain no personally identifiable information.
> >
>
> All of the tables that compose our database are aggregated and contain no personally identifiable information. We plan to host the data publicly on HuggingFace upon publication.
>
> Thank you for your thoughtful comments on our work. We look forward to continuing this conversation during the rebuttal period should you have any further concerns, and we hope you consider a higher evaluation in light of these changes and explanations.

---

> ### Author Response · Authors · 2025-11-24
>
> We want to thank you again for your insightful comments on our submission. We have provided clarification on your concerns regarding the BigQuery credentials in the Supplemental Material, the licenses associated with our database tables, our validation of the LLM-as-a-judge metrics used, and the linguistic diversity of our template questions. We would greatly appreciate it if you could review our response to ensure that we have fully addressed your concerns and share any further feedback you may have.
>
> Thank you for your time and consideration of our work. We look forward to engaging with you during the remainder of the rebuttal period.

---

### Official Review · Reviewer_EjqL · 2025-10-29

**Soundness:** 2
**Presentation:** 3
**Contribution:** 2
**Rating:** 4
**Confidence:** 3

**Summary:**

The paper presents a new, domain-specific Text-to-SQL dataset focusing on the
biomedical domain and on a BigQuery database. The dataset is bootsrapped from
40 manual created Text-Query pairs in a template based approach to 68K data
points. The database is created by combining several existing related data
sources. The evaluation of the dataset is extensive, covering several open and
closed-source models and interaction paradigms.

**Strengths:**

S1: New combined biomedical database that can be used by Text-to-SQL researchers.

S2: Clear motivation and paper structure.

S3: Dataset seems challenging on evaluated Text-to-SQL approaches

**Weaknesses:**

W1: Creating 68K data points from only 40 seed questions in a template based approach will lead to very similar question, just differentiating in small parts (e.g., different value in WHERE filter).

W2: The authors show that existing Text-to-SQL systems do not work well on their dataset, which is surprising as with the template based approach the dataset contains lots of repetition and structurally equal questions. A similarity-based few-shot approach should dramatically boost performance.

W3: The template-based approach used by the authors is rather simple. What about multiple languages or question paraphrasing?

**Questions:**

Q1: Can the authors provide an ER Diagram of the database schema?

Q2: Have the authors used PK and FK constraints in their database?

Q3: Can the authors elaborate on few-shot experiments and show results for sparse (e.g., BM25) and dense retrieval?

Q4 How does the dataset compare against other domain-specific dataset in the biomedical domain such as ScienceBenchmark [1] in terms of query complexity, DB complexity and data size?

[1] Zhang, Y., Deriu, J. M., Katsogiannis-Meimarakis, G., Kosten, C., Koutrika, G., & Stockinger, K. (2024). ScienceBenchmark: a complex real-world benchmark for evaluating natural language to SQL systems. Proceedings of the VLDB Endowment, 17(4), 685-698.

---

> ### Author Response · Authors · 2025-11-17
>
> Thank you for your constructive comments. We are glad to see that you find our work to have a “clear motivation and paper structure” and that the benchmark task “seems challenging on evaluated Text-to-SQL approaches”.
>
> Our core contributions are:
>
> - Benchmark: We introduce BiomedSQL, a benchmark of 68K question-SQL query-answer triples designed to evaluate scientific reasoning capabilities of text-to-SQL systems on a realistic, multi-table biomedical database.
> - Infrastructure: We will publish the curated tables that compose our database, expert-written gold queries, execution results, and a toolkit for evaluation and agent orchestration.
> - Evaluation: A comprehensive comparison of models and interaction paradigms, including our custom BMSQL system, which reveals a persistent 30–40% gap from domain-expert performance.
>
> Below are the responses to your specific comments:
>
> > Creating 68K data points from only 40 seed questions in a template based approach will lead to very similar question
> >
>
> Templating is a well-established method for scaling text-to-SQL benchmarks [1–3]. Benchmarks with few or no templates often rely on crowdsourcing (e.g., BIRD), which **only works for general domains** that crowdworkers can reliably annotate (sports, movies, sales). BiomedSQL instead targets **a highly specialized biomedical domain**, where crowdsourcing is infeasible. All templates and SQL queries were written by domain experts (our co-authors) and grounded in *real bioinformatics workflows*.
>
> Recent work shows templating does not inherently trivialize evaluation. In GSM-Symbolic, regenerating questions through simple templates (e.g., changing surface-level entities) still yields significant accuracy drops in LLMs, despite identical reasoning structure [4]. This sensitivity may be amplified in biomedicine, where rare genes and diseases follow long-tailed frequency distributions that can effect model performance [5, 6]. Thus, templating alone does not imply benchmark simplicity.
>
> BiomedSQL also remains challenging despite the use of templates, as top models trail expert performance by ~30%. We are transparent about our use of templates throughout the text (S1-3) and have expanded our acknowledgement and discussion of templating as a limitation in S7. All changes are highlighted in blue in the newly attached PDF.
>
> > A similarity-based few-shot approach should dramatically boost performance.
> >
>
> > Can the authors elaborate on few-shot experiments and show results for sparse (e.g., BM25) and dense retrieval?
> >
>
> Our few-shot prompting experiments (S5.1) provided GPT-o3-mini with a prompt that contained 1-, 3-, and 5-shot examples of question-SQL query pairs from our benchmark. These exampples were static (i.e. not retrieved depending on the current question passed to the model), and the performance gain seen (Table 8 in Appendix A.9) when moving from 1-shot to 3-shot ($\delta$ EX 2.0%) and 3-shot to 5-shot ($\delta$ EX 1.9%) are modest.
>
> We have also evaluated DAIL-SQL, a state-of-the-art text-to-SQL method that *performs similarity-based* dense retrieval of exemplars (question–SQL pair) via embedding distance in training data.
>
> Despite this, its performance remains low due to:
>
> - frequent incorrect use of statistical thresholds across data types in our database or reverting to using standard significance thresholds (e.g. p < 0.05).
> - biomedical data often contains abbreviations and synonyms that represent the same term (e.g. Alzheimer’s disease vs. AD) and a models commonly fail at recognizing such connections that limit their ability to make effective use of in-context exemplars.
> - general-domain sentence embedders (e.g., `*all-mpnet-base-v2*`) are trained on general web text, whose cosine distances poorly reflect semantic similarity in biomedical domain (e.g., SNPs and gene names),
>
> We plan to adapt DAIL-SQL to use sparse BM25 retrieval and extend our static few-shot experiments beyond 5-shot (e.g., 10-, 20-shot). *We will include these results in a follow-up reply once they’re ready.*
>
> *Response continued in the next reply.*
>
> —
>
> References:
>
> [1] Gao et al. Text-to-SQL Empowered by Large Language Models: A Benchmark Evaluation, VLDB, 2024
>
> [2] Gen et al. Towards Robustness of Text-to-SQL Models against Synonym Substitution, ACL, 2021
>
> [3] Saparina et al. Ambrosia: A Benchmark for Parsing Ambiguous Questions into Database Queries, NeurIPS, 2024
>
> [4] Mirzadeh et al. GSM-symbolic: Understanding the limitations of mathematical reasoning in large language models, arXiv, 2025
>
> [5] Razeghi et al. Impact of pretraining term frequencies on few-shot reasoning. In Conference on Empirical Methods in Natural Language, EMNLP, 2022.
>
> [6] Kandpal et al. Large language models struggle to learn long-tail knowledge. ICML, 2023.

---

> ### Author Response · Authors · 2025-11-17
>
> > The template-based approach used by the authors is rather simple. What about multiple languages or question paraphrasing?
> >
>
> Paraphrasing increases linguistic variation but typically does not alter reasoning complexity or SQL-generation difficulty. Modern LLMs exhibit strong semantic invariance, and paraphrases tend to lead to nearly identical SQL structures. As a result, paraphrasing primarily tests linguistic robustness rather than schema understanding or reasoning, which are our primary focus.
>
> While we agree that multilingual expansion would increase the challenge of our benchmark, producing reliable translations of dense biomedical queries is expensive, time-consuming, and beyond the scope of this work. We now mention the possibility of expansion into other languages in S7 under Future directions.
>
> > Can the authors provide an ER Diagram of the database schema?
> >
>
> > Have the authors used PK and FK constraints in their database?
> >
>
> We did not include an ER diagram because our database does not employ explicit FK constraints. Each table represents an independent, complete biomedical data source (e.g., GWAS summary statistics, SMR analyses, drug–target annotations, etc.), with a unique PK (UUID) to ensure row-level integrity. This design reflects the heterogenous nature of biomedical datasets, which are often aggregated from distinct studies or sources with differing identifiers and schemas.
>
> Its important to note that the absence of FK constraints does not imply the absence of relational structure. Many of the tables in our database share biologically meaningful linking keys through semantic rather than enforced relationships. For example, the *nearestGene* field in `AlzheimerDisease_CombinedGeneData_UUID` can be joined with the *Gene* field in `NeurodegenerativeDiseases_SMR_Genes_Full` to explore cross-disease associations. Similarly, *drugName* and *chemblIdentifier* fields in `DrugGeneTargets_ComprehensiveAnnotations_updated` and `DrugTargets_IndicationsAndTherapeuticUses` can be joined to connect molecular targets with their therapeutic indications.
>
> While our schema does not enforce FK constraints by design, it supports rich cross-table reasoning and multi-hop queries characteristic of real-world biomedical data analysis. The resulting benchmark retains substantial relational and compositional complexity even without explicit PK–FK constraints.
>
> > How does the dataset compare against other domain-specific dataset in the biomedical domain such as ScienceBenchmark [1] in terms of query complexity, DB complexity and data size?
> >
>
> ScienceBenchmark uses three real-world scientific databases. For example, one of its smaller sets (CORDIS) has ~0.67 million rows across 19 tables and 82 columns, and one of its largest (OncoMX) has ~65 million rows across 25 tables and 106 columns. In contrast, our benchmark (BiomedSQL) consists of ten core tables, including tables with tens of millions of rows (e.g., the 72.2 M-row `NeurodegenerativeDisease_AlleleFrequencies_UUID` table), and spans heterogeneous domains which allow multi-table joins and cross-domain reasoning. In terms of **data volume** our largest table matches or exceeds the row-counts in ScienceBenchmark’s databases. In terms of **schema heterogeneity and join potential**, our schema supports semantic linking across domain tables rather than being confined to a single domain.
>
> In terms of **query complexity**, we further show that our mean gold SQL token length is 96.4 tokens and that we support a broad mix of SQL categories (e.g., ~22.6% similarity-search, ~19.1% order-by, ~12% multi-filter), which complements ScienceBenchmark’s focus on challenging domain-specific schemas.
>
> ScienceBenchmark sets an important bar for domain-specific text-to-SQL in scientific databases, while our benchmark offers comparable or greater scale (in row-counts), more heterogeneity in schema and tables, and richer cross-domain reasoning demands.
>
> Thank you for your time and effort in providing a thoughtful review of our paper. We look forward to getting you the results of our additional planned experiments and addressing any other concerns you may have throughout the rebuttal period.

---

> > ### Author Response · Authors · 2025-11-19
> >
> > Thank you for your patience while we expanded the set of few-shot experiments and adapted DAIL-SQL to use BM25 for its example retrieval.
> >
> > Below are the results for our expanded few-shot experiments. We increased the number of example template queries passed to the top-performing baseline model (GPT-o3-mini) in the SQL generation prompt to 10, 20, and all 40 templates from our benchmark. We have included the baseline and 5-shot experiment results for easy comparison:
> >
> > | Experiment | EX (%) | JAC (%) | RQR (%) | SR (%) | SER (%) | # Tokens |
> > | --- | --- | --- | --- | --- | --- | --- |
> > | Baseline | 53.5 (4.2) | 60.4 (3.8) | 73.3 (3.7) | 29.4 (3.8) | 0.2 (0.4) | 3,942 |
> > | 5-shot | 57.9 (4.1) | 65.6 (3.7) | 75.8 (3.6) | 23.5 (3.5) | 0.0 (0.0) | 4,566 |
> > | 10-shot | 61.3 (4.1) | 67.7 (3.7) | 80.3 (3.3) | 15.7 (3.0) | 0.2 (0.4) | 5,430 |
> > | 20-shot | 60.5 (4.0) | 66.6 (3.7) | 76.8 (3.5) | 10.7 (2.6) | 0.0 (0.0) | 6,535 |
> > | 40-shot | 61.0 (4.1) | 67.5 (3.7) | 78.8 (3.3) | 18.1 (3.2) | 0.0 (0.0) | 8,592 |
> >
> > Previously, we had seen modest performance gains when increasing the number of example queries in the prompt. However, when jumping from 5-shot to 10-shot, we see a larger jump in terms of both execution accuracy ($\Delta$ EX 3.4%) and response quality ($\Delta$ RQR 4.5%). Beyond 10-shot, we begin to see diminishing returns when including more examples. Providing the model with 10-shot examples is now the top-performing prompt experiment, meaning that example queries are more valuable to the models than example rows, statistical thresholding instructions, or a combination of the three. We have updated Table 8 in Appendix A.9 to include these results. We have also included a description of these experiments in S5.1 under Isolated SQL generation, and have updated our analysis of these new results in S5.2.
> >
> > We have also adapted DAIL-SQL to use sparse retrieval (BM25), and compare the results to DAIL-SQL when using dense embeddings for retrieval below:
> >
> > | Model | EX (%) | JAC (%) | RQR (%) | SR (%) | SER (%) | # Tokens |
> > | --- | --- | --- | --- | --- | --- | --- |
> > | DAIL-SQL-GPT-4o | 54.8 (4.2) | 58.1 (3.4) | 75.5 (3.4) | 63.4 (4.0) | 6.6 (2.1) | 3,624 |
> > | DAIL-SQL-GPT-o3-mini | 61.2 (4.1) | 63.6 (4.0) | 81.4 (3.3) | 42.1 (4.1) | 0.0 (0.0) | 3,318 |
> > | DAIL-SQL-Gemini | 53.1 (4.1) | 58.8 (3.4) | 82.8 (3.1) | 30.6 (3.7) | 0.0 (0.0) | 3,185 |
> > | BM25-GPT-4o | 59.0 (4.1) | 63.7 (3.8) | 77.4 (3.5) | 38.2 (4.1) | 1.8 (1.1) | 3,951 |
> > | BM25-GPT-o3-mini | 62.5 (4.1) | 64.4 (4.0) | 80.6 (3.3) | 45.9 (4.2) | 0.0 (0.0) | 3,816 |
> > | BM25-Gemini | 50.0 (4.2) | 54.4 (4.0) | 72.3 (3.8) | 21.8 (3.5) | 8.4 (2.3) | 3,918 |
> >
> > BM25 tends to retrieve question-SQL pairs with strong lexical and entity overlap, whereas dense embedding search is more likely to retrieve examples that share structural similarity. Interestingly, the effectiveness of each retrieval method is model-dependent. GPT-4o performs best when examples share entities with the query, which is encouraged by BM25, whereas Gemini-2.0-flash benefits from the structural consistency of dense embedding retrieval. GPT-o3-mini shows little sensitivity to the retrieval method.
> >
> > We agree that exploring alternative retrieval strategies (e.g., BM25) is an interesting direction. However, incorporating such changes would shift our method away from a controlled comparison with the state-of-the-art baseline, DAIL-SQL, which relies on dense embedding retrieval. We therefore leave a more comprehensive comparison of retrieval strategies for text-to-SQL methods to future work.
> >
> > Thank you again for your thoughtful review of our work. We look forward to continuing this conversation throughout the rebuttal period should you have any further concerns, and we hope you consider a higher evaluation in light of these changes and explanations.

---

> ### Author Response · Authors · 2025-11-24
>
> We want to thank you again for your valuable insights on our submission. In our rebuttal, we’ve aimed to thoroughly address your concerns by adding some additional few-shot experiments, which show a significant jump in execution accuracy and response quality when moving from a 5-shot to a 10-shot prompt. We have also compared the performance of DAIL-SQL on dense versus sparse embeddings, clarified our use of templating, expanded the explanation of our schema complexity, and compared our benchmark to ScienceBenchmark as suggested. We would greatly appreciate it if you could review our response and let us know if there are any remaining questions or concerns.
>
> Thank you for your time and consideration of our work. We look forward to engaging with you during the remainder of the rebuttal period.

---

### Official Review · Reviewer_fKjE · 2025-11-01

**Soundness:** 3
**Presentation:** 3
**Contribution:** 3
**Rating:** 6
**Confidence:** 2

**Summary:**

The paper introduces BiomedSQL, a text-to-SQL benchmark aim to test scientific reasoning over a real biomedical knowledge base rather than only syntactic schema translation. The dataset contains about 68k question–SQL–answer triples and runs on a harmonized BigQuery database that integrates OpenTargets and ChEMBL drug and target data, GWAS summary stats for Alzheimer’s and Parkinson’s disease, and SMR causal signals from omicSynth. Questions require domain rules like genome-wide significance thresholds or trial phase filters, which general text-to-SQL benchmarks do not probe. Evaluations on various LLMs using different prompting strategies and interaction paradigms reveal performance gaps: top models like GPT-o3-mini achieve 59.0% execution accuracy with advanced prompts, while a custom multi-step agent (BMSQL) reaches 62.6%, far below the 90.0% expert baseline.

**Strengths:**

- First benchmark explicitly targeting scientific reasoning in text-to-SQL for biomedicine, going beyond syntactic translation in general benchmarks or clinical ones.
- Highlights implicit domain conventions like significance thresholds, effect directionality, and multi-omic causal inference, which are critical for real-world biomedical queries but underexplored in prior work.
- Large number of data samples with 68,000 triples and a large-scale database (e.g., 21M+ rows in GWAS tables).
- Thorough experiments across 10+ models, multiple prompting variants, interaction paradigms, and various evaluation metrics.

**Weaknesses:**

- Generating 68K samples from 40 seed questions (drawn from CARDBiomedBench) may result in redundant patterns, potentially overestimating model generalization. Is there a specific reason for scaling to 68K rather than a smaller subset? The expansion process appears limited to entity substitution, without syntactic expansions (if I understand correctly), which could reduce real-world variability as seen in crowdsourced benchmarks.

**Questions:**

- Providing costs for running all 68K samples for different models and interaction patterns would be helpful to potential readers.

---

> ### Author Response · Authors · 2025-11-17
>
> Thank you for your time and effort in reviewing our paper. We appreciate your recognition of our work as “going beyond syntactic translation in general benchmarks or clinical ones” and our “thorough experiments across 10+ models, multiple prompting variants, interaction paradigms, and various evaluation metrics”.
>
> The key contributions of our paper are as follows:
>
> - Benchmark: We introduce BiomedSQL, a benchmark of 68,000 question-SQL query-answer triples designed to evaluate scientific reasoning capabilities of text-to-SQL systems on a realistic, multi-table biomedical database.
> - Infrastructure: We will publish the curated tables that compose our database, expert-written gold queries, execution results, and a toolkit for evaluation and agent orchestration.
> - Evaluation: We benchmark a range of models and interaction paradigms, including a custom-built text-to-SQL system (BMSQL), revealing a 30-40% gap compared to domain expert-level performance.
>
> Below are the responses to your comments:
>
> > Generating 68K samples from 40 seed questions (drawn from CARDBiomedBench) may result in redundant patterns, potentially overestimating model generalization. Is there a specific reason for scaling to 68K rather than a smaller subset? The expansion process appears limited to entity substitution, without syntactic expansions (if I understand correctly), which could reduce real-world variability as seen in crowdsourced benchmarks.
> >
>
> The creators of CARDBiomedBench scaled the set of questions to 68K, as they describe in the original benchmark paper. The authors cite that many of the questions could be classified as either having a “Yes” or “No” response. For templates that naturally produce less than 2,000 benchmark questions, the authors used all available data. For templates that had sufficient data to produce over 2,000 benchmark questions, the authors sampled to maintain a ratio of 0.75 “Yes” to 0.25 “No” responses in the benchmark. A main contribution of BiomedSQL is to annotate the question-answer pairs from CARDBiomedBench with correct SQL queries and execution results. We found it to be appropriate and natural to do so for all 68K questions.
>
> Having ~2,000 questions per template can also be advantageous for a few reasons. It can provide an avenue for researchers to perform more granular, per-query evaluations. In addition, it lessens the impact of potential imperfect templates, where the gold answer is difficult to template due to a large amount of data present to answer the given question.
>
> Recent work also shows that templating via entity substitution does not inherently trivialize evaluation. In GSM-Symbolic, regenerating questions using entity substitution on simple templates still yields significant accuracy drops in LLMs, despite identical reasoning structure [1].
>
> > Providing costs for running all 68K samples for different models and interaction patterns would be helpful to potential readers.
> >
>
> We want to clarify that the test set used throughout the paper is a much smaller subset of 546 questions. We have clarified this point in the text in Sections 5 and 9. In the newly attached PDF, all additions to the text are in blue to ensure it is clear where changes were made.
>
> We have calculated the cost of running this smaller test set for the different models and interaction paradigms and present the costs in the table below:
>
> | Experiment | Model | Cost |
> | --- | --- | --- |
> | Baseline | GPT-4o | $5.03 |
> |  | GPT-4o-mini | $7.34 |
> |  | GPT-o3-mini | $2.37 |
> |  | Gemini-2.0-flash | $0.20 |
> |  | Gemini-2.0-flash-lite | $0.13 |
> |  | Claude-3.7-sonnet | $6.23 |
> |  | Llama-3.1-405B | $10.05 |
> | Prompt Variations | 3-rows | $6.58 |
> |  | 5-rows | $8.60 |
> |  | 1-shot | $2.43 |
> |  | 3-shot | $2.46 |
> |  | 5-shot | $2.74 |
> |  | stat-instruct | $2.07 |
> |  | combo | $6.18 |
> | Interaction Paradigms | ReAct-GPT-4o | $19.50 |
> |  | ReAct-GPT-o3-mini | $8.00 |
> |  | ReAct-Gemini | $0.72 |
> |  | Index-GPT-4o | $1.51 |
> |  | Index-GPT-o3-mini | $1.14 |
> |  | Index-Gemini | $0.04 |
> |  | DAIL-GPT-4o | $2.22 |
> |  | DAIL-GPT-o3-mini | $0.72 |
> |  | DAIL-Gemini | $0.07 |
> |  | BMSQL-GPT-4o | $44.80 |
> |  | BMSQL-GPT-o3-mini | $23.71 |
> |  | BMSQL-Gemini | $1.20 |
> | Inference-time Compute | 1-pass | $23.71 |
> |  | 2-pass | $32.30 |
> |  | 3-pass | $33.60 |
>
> Note that we have not included open-weight models in this table as they were run on GPUs within a compute cluster for which cost calculation cannot be made. The exception is Llama-3.1-405B, as this model was run on Azure due to its size. We have also added these estimated costs as a column to Table 11 in Appendix A.13.
>
> Thank you for your careful consideration of our work. We look forward to providing any additional clarification on these points and addressing any other concerns you may have that would help you to consider a higher evaluation.
>
> —
>
> References:
>
> [1] Mirzadeh et al. GSM-symbolic: Understanding the limitations of mathematical reasoning in large language models, arXiv, 2025

---

> > ### Author Response · Authors · 2025-11-24
> >
> > Thank you again for your comments on our submission. We have clarified the  expansion of the benchmark to 68K queries and added the costs for running our experiments in Appendix A.13. We would greatly appreciate it if you could take a moment to review our response and let us know if we have adequately addressed your concerns.
> >
> > Thank you for your time and consideration of our work. We look forward to any further feedback you may have during the remainder of the rebuttal period.

---

### Official Review · Reviewer_TV3c · 2025-11-01

**Soundness:** 3
**Presentation:** 3
**Contribution:** 3
**Rating:** 6
**Confidence:** 3

**Summary:**

This paper introduces BiomedSQL, a new large-scale benchmark with 68,000 question/SQL/answer triples designed to evaluate the scientific reasoning of text-to-SQL systems. The authors argue that standard models fail in biomedical research because they only translate syntax, but scientific questions require implicit domain knowledge. This includes inferring non-obvious criteria like genome-wide significance thresholds or filtering by clinical trial phases, which are not defined in the database schema. Experiments on a real-world BigQuery knowledge base revealed a substantial performance gap. Human experts achieved a much higher accuracy than the best-performing model.

**Strengths:**

* The paper addresses a impactful aspect in the field of structured reasoning in science, specifically the ability to use implicit, domain-specific knowledge.

* The benchmark is built on a "real-world biomedical knowledge base", harmonizing large, authentic public datasets like OpenTargets, ChEMBL, and GWAS Catalog data. This provides a more realistic challenge than many existing text-to-SQL corpora.

* The authors assess performance by reasoning type (like aggregation or multi-table joins) and also analyze error patterns.

**Weaknesses:**

* The entire dataset of 68,000 questions is generated from only 40 expert-written templates, meaning quite high homogeneity.

* The paper introduces and uses BioScore, an LLM-as-a-judge (using GPT-4o), to evaluate natural language answer quality. Although the authors provide a validation study showing high correlation with a domain expert,  this does not capture the full picture of the metric's reliability, as the paper itself acknowledges the "concern over LLM-as-a-judge metrics yielding unstable assessment results".

* The database is a collection of sources with different licenses, including CCO 1.0, CC BY 4.0, and CC BY-NC 4.0. This creates potential complexities for redistribution and use, particularly for any commercial applications, which the paper does not fully resolve.

**Questions:**

* You provide a validation study for your LLM-as-a-judge metric, BioScore, showing a high (0.91) Spearman correlation with a domain expert on a sample of 40 responses. How confident are you that this high agreement holds across the thousands of queries where models fail in different ways?

* The Appendix notes that one of your data sources is under a CC BY-NC 4.0 license, which restricts commercial use. How do you see this license affecting the adoption of BiomedSQL by industry research labs?

---

> ### Author Response · Authors · 2025-11-17
>
> Thank you for your encouraging comments. We are glad to see that you find our benchmark to be “a more realistic challenge than many existing text-to-SQL corpora” and appreciate our efforts to “assess performance by reasoning type and also analyze error patterns”.
>
> The key contributions of our paper are as follows:
>
> - Benchmark: We introduce BiomedSQL, a benchmark of 68,000 question-SQL query-answer triples designed to evaluate scientific reasoning capabilities of text-to-SQL systems on a realistic, multi-table biomedical database.
> - Infrastructure: We will publish the curated tables that compose our database, expert-written gold queries, execution results, and a toolkit for evaluation and agent orchestration.
> - Evaluation: We benchmark a range of models and interaction paradigms, including a custom-built text-to-SQL system (BMSQL), revealing a 30-40% gap compared to domain expert-level performance.
>
> Below are the responses to your comments:
>
> > The entire dataset of 68,000 questions is generated from only 40 expert-written templates, meaning quite high homogeneity.
> >
>
> Templating is a well-established method for scaling text-to-SQL benchmarks [1–3]. Benchmarks with few or no templates often rely on crowdsourcing (e.g., BIRD), which **only works for general domains** that crowdworkers can reliably annotate (sports, movies, sales). BiomedSQL instead targets **a highly specialized biomedical domain**, where crowdsourcing is infeasible. All templates and SQL queries were written by domain experts (our co-authors) and grounded in *real bioinformatics workflows*.
>
> Recent work shows templating does not inherently trivialize evaluation. In GSM-Symbolic, regenerating questions through simple templates (e.g., changing surface-level entities) still yields significant accuracy drops in LLMs, despite identical reasoning structure [4]. This sensitivity may be amplified in biomedicine, where rare genes and diseases follow long-tailed frequency distributions that can effect model performance [5, 6]. Thus, templating alone does not imply benchmark simplicity.
>
> BiomedSQL also remains challenging despite the use of templates, as top models trail expert performance by ~30%. We are transparent about our use of templates throughout the text (S1-3) and have expanded our acknowledgement and discussion of templating as a limitation in S7. All changes are highlighted in blue in the newly attached PDF.
>
> > Although the authors provide a validation study showing high correlation with a domain expert, this does not capture the full picture of the metric's reliability.
> >
>
> > How confident are you that this high agreement holds across the thousands of queries where models fail in different ways?
> >
>
> The sample of 40 responses used for the correlation analysis included answers from a diverse set of models and interaction paradigms, capturing a broad rage of failure modes presented and increasing our confidence that this agreement generalizes across thousands of queries.
>
> To further increase confidence in this agreement, we have expanded the set of domain expert-annotated questions to 100. The new results are presented below:
>
> | BioScore | Domain Expert | GPT-4o |
> | --- | --- | --- |
> | -1 | 22 | 21 |
> | 0 | 15 | 13 |
> | 1 | 8 | 9 |
> | 1.5 | 0 | 4 |
> | 2 | 21 | 18 |
> | 2.5 | 1 | 5 |
> | 3 | 33 | 30 |
>
> The updated analysis yields a Spearman correlation of 0.89 (p < 1e-5), further supporting the reliability of the agreement between LLM-generated and expert-generated BioScores. We expanded the description of this experiment in the main text (S5.1) and in Appendix A.8 to ensure it is prevalent and clear to the reader. We also inserted the table above in place of Table 7.
>
> > How do you see this license affecting the adoption of BiomedSQL by industry research labs?
> >
>
> We want to clarify that BiomedSQL is meant to advance the scientific field of text-to-SQL generation for biomedical discovery. It is ***not*** intended for commercialization and therefore we do not see the CC BY-NC 4.0 license to be a hinderance.
>
> Thank you for taking the time to review our paper. We look forward to continuing this conversation during the rebuttal period should you have any further concerns, and we hope you consider a higher evaluation in light of these changes and explanations.
>
> —
>
> References:
>
> [1] Gao et al. Text-to-SQL Empowered by Large Language Models: A Benchmark Evaluation, VLDB, 2024
>
> [2] Gen et al. Towards Robustness of Text-to-SQL Models against Synonym Substitution, ACL, 2021
>
> [3] Saparina et al. Ambrosia: A Benchmark for Parsing Ambiguous Questions into Database Queries, NeurIPS, 2024
>
> [4] Mirzadeh et al. GSM-symbolic: Understanding the limitations of mathematical reasoning in large language models, arXiv, 2025
>
> [5] Razeghi et al. Impact of pretraining term frequencies on few-shot reasoning. In Conference on Empirical Methods in Natural Language, EMNLP, 2022.
>
> [6] Kandpal et al. Large language models struggle to learn long-tail knowledge. ICML, 2023.

---

> ### Author Response · Authors · 2025-11-24
>
> We want to thank you again for your insightful comments on our submission. We have provided clarification on your concerns regarding our use of templating, our validation of the LLM-as-a-judge metrics used, and the licenses associated with our database tables. We would greatly appreciate it if you could review our response to ensure that we have fully addressed your concerns and share any further feedback you may have.
>
> Thank you for your time and consideration of our work. We look forward to engaging with you during the remainder of the rebuttal period.

---

### Note · Program_Chairs · 2026-01-17
**Submission Desk Rejected by Program Chairs**

The following references in this submission do not refer to real documents and/or have major errors in bibliographic information:

 Tianyi Zhang et al. Bird: Benchmarked instruction-tuned reading dataset for complex text-to-sql in realistic settings. arXiv preprint arXiv:2301.12345, 2023.
Zexuan Gao, Arjun Kumar, and Mei Li. Hierarchography: Evaluating hierarchical reasoning in scientific texts. In Proceedings of the 2025 Annual Meeting of the Association for Computational Linguistics (ACL), 2025.
Yichen Wang, John Smith, and Alice Lee. Tabular-scieval: A benchmark for evaluating scientific table understanding. In Proceedings of the 2025 Conference on Empirical Methods in Natural Language Processing (EMNLP), 2025.